# RMAAT: Astrocyte-Inspired Memory Compression and Replay for Efficient Long-Context Transformers

**Md Zesun Ahmed Mia, Malyaban Bal & Abhronil Sengupta** [*]
School of Electrical Engineering and Computer Science
Pennsylvania State University
University Park, PA 16802, USA
{zesun.ahmed,mjb7906,sengupta}@psu.edu

## Abstract

The quadratic complexity of self-attention mechanism presents a significant impediment to applying Transformer models to long sequences. This work explores computational principles derived from astrocytes—glial cells critical for biological memory and synaptic modulation—as a complementary approach to conventional architectural modifications for efficient self-attention. We introduce the Recurrent Memory Augmented Astromorphic Transformer (RMAAT), an architecture integrating abstracted astrocyte functionalities. RMAAT employs a recurrent, segment-based processing strategy where persistent memory tokens propagate contextual information. An adaptive compression mechanism, governed by a novel retention factor derived from simulated astrocyte long-term plasticity (LTP), modulates these tokens. Attention within segments utilizes an efficient, linear-complexity mechanism inspired by astrocyte short-term plasticity (STP). Training is performed using Astrocytic Memory Replay Backpropagation (AMRB), a novel algorithm designed for memory efficiency in recurrent networks. Evaluations on the Long Range Arena (LRA) benchmark demonstrate RMAAT's competitive accuracy and substantial improvements in computational and memory efficiency, indicating the potential of incorporating astrocyte-inspired dynamics into scalable sequence models.

## 1 Introduction

The Transformer architecture (Vaswani et al., 2017) has become foundational for sequence modeling, particularly in natural language processing. A primary limitation, however, is the quadratic computational and memory complexity ($O(N^2)$) of its self-attention mechanism, hindering its application to very long sequences (Tay et al., 2020; Beltagy et al., 2020). The predominant research direction to overcome this focuses on modifying the Transformer architecture itself for greater efficiency. Techniques explored include sparse attention patterns (Child et al., 2019; Beltagy et al., 2020), linear attention approximations (Katharopoulos et al., 2020; Peng et al., 2021), state-space models (Gu et al., 2021; Gu & Dao, 2023), and various recurrent structures (Peng et al., 2023; Sun et al., 2023; Yang et al., 2023; Bulatov et al., 2022). Alongside efforts to improve architectural efficiency, research into brain-inspired computational principles is gaining interest, driven by the potential for remarkable energy efficiency and novel processing mechanisms. However, similar to the challenges faced by conventional architectures, developing neuro-inspired learning approaches that robustly handle complex, long-range dependencies while being both computationally efficient and biologically grounded remains a significant hurdle (Bal & Sengupta, 2024). Addressing this challenge may require looking beyond purely neuronal models, as many brain-inspired computing approaches focus predominantly on neuronal activity, often overlooking the computational roles of other critical cell types.

---

[*]Code Available: https://github.com/NeuroCompLab-psu/RMAAT.git

Among these overlooked elements are astrocytes, a type of glial cell increasingly recognized not just for support functions but for their active participation in modulating synaptic transmission, plasticity, and memory processes critical for learning (Gibbs et al., 2008; Bohmbach et al., 2022; Perea et al., 2009; Alberini et al., 2018). Given their established role in modulating temporal information and memory consolidation within biological circuits, we build on the premise that principles derived from astrocyte function are particularly well-suited to addressing the long-range temporal dependency challenges inherent in processing extended sequences. Despite their potential, astrocyte-based computational principles remain severely underexplored in deep learning.

This paper introduces the Recurrent Memory Augmented Astromorphic Transformer (**RMAAT**), an architecture that integrates specific, computationally abstracted astrocyte-inspired mechanisms related to temporal memory processing (inspired by astrocyte long-term effects) and attention modulation (inspired by astrocyte short-term effects) within a recurrent transformer framework. Our goal is to leverage these neuro-glial principles to create an efficient approach for long-context sequence processing. Within this emerging line of work, foundational studies have shown that tripartite synapses can implement Transformer self-attention, validated via weights extracted from pre-trained networks (Kozachkov et al., 2023), and have developed theoretical models of neuron–astrocyte associative memory and capacity scaling (Kozachkov et al., 2025). Subsequent Astromorphic Transformer architectures (Mia et al., 2025) have shown that these principles can be instantiated in standard-scale machine learning models. Building on this trajectory, our work scales these astrocyte-inspired mechanisms to a demanding long-context benchmark (LRA), where they yield competitive performance and improved memory efficiency in the long-sequence regime. The remainder of this paper is organized as follows: Section 2 details our main contributions and positions RMAAT relative to prior work. Section 3 describes the RMAAT model architecture and its bio-inspired components. Section 4 presents experiments and results. Section 5 discusses limitations and concludes the paper.

## 2 RELATED WORKS AND MAIN CONTRIBUTIONS

Significant research addresses the ($O(N^2)$) complexity and long-context limitations of standard Transformers (Vaswani et al., 2017). Early efficiency improvement efforts focused on sparse or linear attention approximations (e.g., Longformer (Beltagy et al., 2020), Reformer (Kitaev et al., 2020)). Others incorporated recurrence via state caching or compression (e.g., Transformer-XL (Dai et al., 2019), Compressive Transformer (Rae et al., 2019)) and some utilized explicit memory tokens to carry context between segments (e.g., RMT (Bulatov et al., 2022), Memformer (Wu et al., 2020)). More recently, highly efficient architectures like State-Space Models (e.g., S4 (Gu et al., 2021), Mamba (Gu & Dao, 2023)) based on continuous-time systems, and RNN/Transformer hybrids (e.g., RetNet (Sun et al., 2023), RWKV (Peng et al., 2023), GLA (Yang et al., 2023)) employing innovations like retention mechanisms or gating, have achieved strong results through sophisticated architectural and mathematical advancements. However, developing methods that integrate deeper biological principles, particularly for complex functions like long-term memory integration, alongside computational efficiency remains an ongoing challenge. Separately, within biologically-inspired computing, astromorphic approaches have explored leveraging astrocyte principles (Kozachkov et al., 2023), particularly adapting attention mechanisms based on astrocyte nonlinearities and inherent plasticities (Mia et al., 2025). While valuable, these efforts have primarily concentrated on the attention component itself. The potential for utilizing computational principles derived from astrocyte temporal dynamics, such as those involved in long-term plasticity (LTP) related to memory formation and consolidation, to specifically address the challenge of long-range context propagation in sequence models remains largely unexplored. Complementary lines of work have investigated neuromodulated Hebbian plasticity in RNNs (Miconi et al., 2020; Duan et al., 2023); in contrast, our model fixes the qualitative modulation structure and timescales from a neuron–astrocyte network simulation and only learns the downstream architectural parameters. To address the above-mentioned gaps, the main contributions of our work are:

**(i) A Distilled Computational Macro Model:** We propose and utilize a novel macro model, distilled from detailed computational models of neuron-astrocyte LTP dynamics (Perea et al., 2009; Alberini et al., 2018), which serves as the foundation for RMAAT's recurrent memory system. **(ii) Memory Retention Factor for Segment-Based Processing:** Building on the macro model (i), we derive a novel **Memory Retention Factor** that bridges the biological abstraction to RMAAT's recurrent architecture, instantiating the macro model's saturation dynamics as a concrete compression schedule for segment-based processing of memory tokens. This factor achieves biologically-

motivated context compression, differing significantly from architectures reliant on externally managed memory (Bulatov et al., 2022; Wu et al., 2020). **(iii) Efficient AMRB Training Algorithm:** We propose the Astrocytic Memory Replay Backpropagation (**AMRB**) algorithm, enabled by the model's memory structure, which significantly reduces the memory footprint and computational overhead compared to standard BPTT or chunk-based backpropagation for recurrent training.

## 3 THE RMAAT MODEL

### 3.1 FOUNDATIONAL COMPUTATIONAL NEUROSCIENCE MODEL

RMAAT's core mechanisms are derived from computational models of the tripartite synapse (Bohmbach et al., 2022; Perea et al., 2009; Alberini et al., 2018), describing neuron-astrocyte interactions. We model key plasticity dynamics operating at different timescales, abstracting the principles into our framework.

**Short-Term Plasticity (STP):** To capture rapid synaptic adjustments and spatial context, we model synaptic facilitation ($s_{ij}$) between a postsynaptic neuron $i$ and a presynaptic neuron $j$, and the associated short-term astrocyte process parameter ($p_{ij}^s$). Their dynamics are conceptually governed by interactions reflecting neuronal co-activation ($\theta(x_i)\theta(x_j)$), astrocyte modulation ($\psi(p_{ij}^s)$), decay ($\beta, \gamma^s$), and coupling between astrocyte processes, operating on a faster timescale ($\tau_s, \tau_p^s$). Simplified representations highlighting key dependencies are:

$$\tau_s \frac{ds_{ij}}{dt} \propto -\beta s_{ij} + \theta(x_i)\theta(x_j) + \psi(p_{ij}^s) \tag{1}$$

$$\tau_p^s \frac{dp_{ij}^s}{dt} \propto -\gamma^s p_{ij}^s + \sum_{k,l=1}^{N} T_{ijkl}\psi(p_{kl}^s) \tag{2}$$

Here, $x_i, x_j$ represent neuronal activity, $\psi(p_{ij}^s)$ represents local astrocyte modulation. In Equation 2, the summation term captures the influence of other astrocyte process activities ($p_{kl}^s$, associated with neuron pairs $k, l$) on the specific process $p_{ij}^s$. The coupling tensor $T_{ijkl}$ represents the concentration fluxes or strength of influence (e.g., via calcium diffusion) between the astrocyte process associated with synapse $(i, j)$ and other processes associated with synapses $(k, l)$. The magnitude of these fluxes typically depends on the relative spatial positions and distances between the interacting synapses within the astrocyte's domain. Thus, the dynamics of $p_{ij}^s$ are modulated by the spatial context encoded in this flux pattern. The influence of these spatially dependent interactions on local astrocyte dynamics provides the biological mapping for how RMAAT computes relative positional information within its attention mechanism (detailed later).

**Long-Term Plasticity (LTP):** To model slower processes related to modulating temporal information and memory consolidation, we consider the long-term astrocyte process parameter ($p_{ij}^l$). This variable integrates the effect of sustained synaptic activity ($s_{ij}$) over a significantly longer timescale ($\tau_p^l > \tau_p^s$), acting as a form of accumulating memory trace.

$$\tau_p^l \frac{dp_{ij}^l}{dt} \propto -\gamma^l p_{ij}^l + \kappa(s_{ij}) \tag{3}$$

The dynamics governed by Equation 3, representing the integration of synaptic history ($s_{ij}$) over longer timescales via the $p_{ij}^l$ variable, provides the conceptual foundation for our subsequent developments. Specifically, we distill the principles captured by these LTP dynamics into a computational **Macro Model** (Contribution 1), which we then operationalize for segment-based sequence processing by deriving a **Memory Retention Factor** (Contribution 2) that instantiates the macro model's saturation curve as a concrete compression schedule for RMAAT's recurrent memory tokens. The detailed derivation, simulation results showing the characteristic behavior, and implementation of this memory system are presented in Section 3.3.

The subsequent sections detail how RMAAT translates these principles into a computational architecture, moving from biophysical simulation to a macro-model of STP/LTP and then to concrete mechanisms for within-segment attention modulation (STP-like) and cross-segment recurrent memory integration (LTP-like) (Debanne & Inglebert, 2023; Bicknell & Latham, 2024; Stasenko & Kazantsev, 2023).

## 3.2 Core Architecture and Processing

RMAAT processes sequences using a recurrent Transformer architecture built upon segmented processing and a bio-inspired attention mechanism with spatial encoding of relative position.

### 3.2.1 Segmented Processing and Biologically Inspired Memory Tokens

To address the quadratic complexity bottleneck of standard self-attention over long sequences, RMAAT adopts a segmented processing approach. The input sequence is divided into non-overlapping, contiguous segments of a manageable maximum length $N_{seg}$. The core RMAAT layers process these segments sequentially, rather than operating on the entire sequence at once. A key element enabling long-range dependency modeling across these segments is the incorporation of dedicated **Memory Tokens**. Inspired by the capacity of biological systems, particularly astrocyte networks, to maintain and integrate information over extended periods (as abstracted in Sec 3.1), these memory tokens serve as a persistent, evolving state. Let the set of $M$ memory tokens at the start of processing segment $t$ be

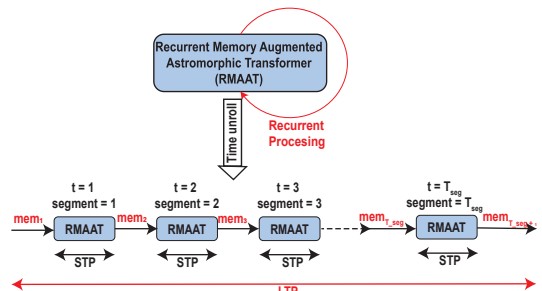

Figure 1: Conceptual illustration of RMAAT processing through time unrolling. Processing within each segment incorporates mechanisms inspired by STP. The recurrent propagation of astrocytic memory tokens ($mem_t$) integrates context across many segments, drawing inspiration from LTP principles for persistent memory.

denoted by $mem_t$. These tokens are processed alongside the actual input tokens $x_t$ within the segment using the mechanisms described below (Sec 3.2.2 and Sec 3.2.3). The output representations corresponding to these memory tokens after processing segment $t$ form the updated memory state, $mem_{t+1}$, which is then passed as the input memory to segment $t + 1$. This recurrent flow, conceptually illustrated in Figure 1, allows contextual information within the memory tokens to propagate across segments. This mechanism differs from approaches like RMT (Bulatov et al., 2022) or Memformer (Wu et al., 2020), which often rely on externally managed memory mechanisms or specific architectural additions for memory updates. In RMAAT, the updates to these memory tokens are intrinsically linked to the bio-inspired dynamics derived from our computational macro model (detailed in Sec 3.3, involving a dynamically derived retention factor), aiming for a more integrated and computationally distinct approach to memory management. The processing within each segment, which updates both sequence and memory token representations, relies on the astromorphic attention mechanism described below.

### 3.2.2 Astromorphic Attention Mechanism

Within each segment processed by RMAAT (as described in Sec 3.2.1), the standard computationally expensive $O(N^2)$ self-attention is replaced by an efficient **Astromorphic Attention** mechanism. Its design draws inspiration from computational models of the tripartite synapse (Mia et al., 2025; Kozachkov et al., 2023) and specifically abstracts principles from the STP dynamics outlined in Section 3.1. To implement this mechanism computationally, we conceptualize it using a two-layer neuron-astrocyte network structure (input/hidden layer and output layer), as depicted abstractly in Figure 2 (Right). The mechanism operates in two consecutive modes within this structure: Write and Read. (See Appendix B for full details).

Let $d$ be the model's embedding dimension (input/output layer size) and $m$ be the hidden layer dimension. For a given segment $t$, the input $X$ consists of the $N_{seq}$ sequence tokens ($x_t$) concatenated with the $M$ memory tokens ($mem_t$), resulting in a total of $N = N_{seq} + M$ tokens processed within the segment. First, the combined input tokens $X \in \mathbb{R}^{N \times d}$ are linearly projected into Keys ($K$), Queries ($Q$), and Values ($V$) using learnable weight matrices $W_K, W_Q \in \mathbb{R}^{d \times m}$ (projecting to the hidden dimension) and $W_V \in \mathbb{R}^{d \times d}$ (projecting to the output dimension), such that $K = XW_K$ (Keys, $\mathbb{R}^{N \times m}$), $Q = XW_Q$ (Queries, $\mathbb{R}^{N \times m}$), and $V = XW_V$ (Values, $\mathbb{R}^{N \times d}$). A non-linear activation function $\phi$ (e.g., $\phi(x) = \text{elu}(x) + 1$, following (Katharopoulos et al., 2020;

Mia et al., 2025)), applied element-wise to $K$ and $Q$, yields activated representations $\phi(K)$ and $\phi(Q)$, analogous to activations in the hidden layer (presynaptic neurons).

The **Write Mode** then computes effective synaptic weights and states within this network structure, encoding context based on Hebbian principles and astrocyte modulation. These represent learned parameters or aggregated states within the network. Specifically: **The Neuronal Hebbian Weight** component ($H_{neuron} \in \mathbb{R}^{m \times d}$) captures the direct correlation between activated keys $\phi(K)$ (hidden layer activations) and values $V$ (output layer), representing baseline Hebbian plasticity summed across the $N$ tokens (conceptually linked to the $\theta(x_i)\theta(x_j)$ co-activation term in Eq. 1).

This models the connection strength between the hidden (presynaptic) and output (postsynaptic) layers based on direct neuron-neuron interaction. **The Astrocyte-Modulated Hebbian Weight** component ($H_{astro} \in \mathbb{R}^{m \times d}$) incorporates the astrocyte's modulatory influence, specifically integrating relative positional information (conceptually linked to the astrocyte modulation term $\psi(p_{ij}^s)$ in Eq. 1, where $p_{ij}^s$ dynamics are spatially modulated). Building on prior astromorphic transformer work (Mia et al., 2025), it uses the activation of a relative positional encoding matrix $R$ (astrocytic parameter), $\phi(R)$, derived from STP dynamics (detailed in Sec 3.2.3) to represent the influence of relative positioning. This models how astrocytes modulate the hidden-to-output layer connection based on spatial context. **Concurrently, the Presynaptic State** ($g \in \mathbb{R}^{1 \times m}$) abstracts the non-linear astrocyte response (e.g., calcium dynamics, denoted by $C \sim Ca^{2+}$ in Figure 2) to the cumulative presynaptic (key) activity $\phi(K)$ within the segment. It aggregates the activated keys $\phi(k_t)$ (where $\phi(k_t)$ is the $t$-th row of $\phi(K)$) over the segment length $N$ and applies a non-linearity controlled by parameter $\alpha$. This state $g$ encodes recent activation history in the hidden layer, influenced by astrocyte dynamics ($\alpha$), relevant for feedback modulation in the Read Mode. These three components, representing learned weights ($H_{neuron}, H_{astro}$) and an aggregated state ($g$) within the network, are calculated as:

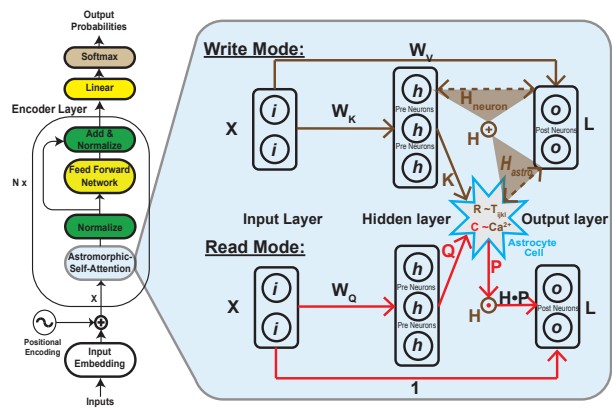

Figure 2: **Overview of the Astromorphic Attention mechanism, detailing the flow from input to output.** This architecture replaces standard self-attention with a bio-inspired, linear-complexity mechanism that emulates the function of a tripartite synapse. The process unfolds in three main steps. **(1) Input Projection:** The input sequence $X$, which includes both the segment's tokens and the recurrent memory tokens, is linearly projected into Key ($K$), Query ($Q$), and Value ($V$) matrices using learnable weight matrices $W_K$, $W_Q$, and $W_V$. **(2) Write Mode (Context Encoding, brown path):** This phase aggregates and encodes contextual information across the entire input segment, modeling how astrocytes integrate signals over a local neighborhood. It computes two Hebbian weight matrices: the **Neuronal Hebbian Weight** ($H_{neuron}$), capturing direct key-value correlations, and the **Astrocyte-Modulated Hebbian Weight** ($H_{astro}$), which is modulated by the astrocytic parameter $R$—a learnable matrix that abstracts the principles of distance-dependent spatial coupling quantified by the tensor $T_{ijkl}$ in the underlying neuroscience model (Sec. 3.2.3)—and introduces spatial context. Together, these Hebbian weights define effective synaptic connections from the hidden layer ($h$) to the output layer ($L$) depicted in the figure. Concurrently, the mechanism computes a presynaptic state $g$, which represents the astrocyte's integrated response to overall neuronal activity and computationally abstracts the dynamics of intracellular calcium concentration ($Ca^{2+}$). **(3) Read Mode (Context Retrieval, red path):** This phase uses the encoded context to generate the final output for each token, modeling astrocyte-mediated feedback. First, each token's unique Query vector is used to compute a dynamic **Presynaptic Plasticity Feedback Factor** ($P$), which acts as a query-specific feedback signal. This factor then modulates the combined Hebbian weights ($H = H_{neuron} + H_{astro}$), and the resulting modulated weight is applied to the queries to produce the final output $L$ at the output layer. The full equations are detailed in Section 3.2.2.

$$H_{neuron} = \frac{1}{m}\phi(K)^T V \qquad H_{astro} = \frac{1}{m}\phi(R)^T V \qquad g = \left(\sum_{t=1}^{N}\phi(k_t)\right)^\alpha \qquad (4)$$

The **Read Mode** uses the current queries $Q$ (projected from input $X$) to retrieve the context encoded during the Write Mode in the combined Hebbian weight $H = H_{neuron} + H_{astro}$. This retrieval is modulated by an astrocyte-inspired feedback mechanism operating within the network structure. First, an interaction strength $C \in \mathbb{R}^{N \times 1}$ between the currently active queries $\phi(Q)$ (hidden layer activations from queries) and the cumulative presynaptic state $g$ (astrocyte state abstraction) is calculated as $C = \phi(Q)g^T$. This represents the calcium response evoked by the presynaptic action potential ($Q$). Inspired by biological modulation (e.g., saturation) (Mia et al., 2025), a **feedback factor** $P$ is derived, typically modeled as inversely related to this interaction strength, i.e., $P = 1/C$. This represents the presynaptic plasticity decoded by the query ($Q$). The combined Hebbian weight matrix $H$ is then modulated element-wise (Hadamard product $\odot$) by this feedback factor $P$. The final weight $H \odot P$ defines the synaptic weight between the hidden ($h$) and output layer ($L$). Activated queries $\phi(Q)$ retrieve the relevant context by multiplying this modulated weight matrix ($H \odot P$). Finally, a standard residual connection adds the original input $X$ to compute the **Final Attention Output** ($L$), which represents the final activation of the output layer for this attention block (further details are provided in Appendix B on how $L$ maps to self-attention in transformer):

$$L = \phi(Q)(H \odot P) + X \qquad (5)$$

The resulting $L \in \mathbb{R}^{N \times d}$ represents the updated token representations for the segment. This computation achieves $O(N)$ complexity because the intermediate context aggregates ($H$ and $g$) are computed once per segment with dimensions independent of $N$, and the final steps involving the $N$-dimensional query matrix $\phi(Q)$ consist of operations like matrix-vector products that scale linearly with $N$, avoiding the quadratic cost of standard attention. The output $L$ typically proceeds through standard subsequent layers like Feed-Forward Networks (FFN) and Layer Normalization within the overall Transformer block structure (Figure 2, left).

### 3.2.3 Biological Grounding of Relative Positional Encoding by STP Dynamics

Effective attention mechanisms in transformer architectures often benefit from incorporating relative positional information to understand sequence order (Shaw et al., 2018; Mia et al., 2025). Common implementations define a base distance matrix—for instance, using an exponential decay $r_{ij} = \exp(-\|\text{pos}_i - \text{pos}_j\| \times \text{scale})$, where $\text{pos}_i$ and $\text{pos}_j$ represent token positions and scale is a tunable hyperparameter controlling the spatial range of influence. This base distance information is then transformed using learnable projections to compute a final positional encoding matrix $R \in \mathbb{R}^{N \times m}$, with specific implementation details discussed in Appendix B and following prior works like (Mia et al., 2025). While such methods are computationally effective, they often lack a direct biological correspondence. Our work seeks to provide this biological grounding by mapping the concept of relative positional encoding to principles observed in simulated astrocyte Short-Term Plasticity (STP) dynamics, particularly the role of the concentration flux tensor $T_{ijkl}$.

Our computational neuroscience simulations (Sec 3.1, Appendix C) investigate spatial interactions among astrocyte processes. These simulations incorporate the distance-dependent coupling tensor $T_{ijkl}$ (Eq. 2), reflecting how influence between processes diminishes with distance, akin to biological signaling like calcium diffusion (Wade et al., 2011; De Pittà et al., 2009) (the $T_{ijkl}$-mediated coupling effectively defines "many-neuron synapses" that couple activity across multiple synapses within an astrocyte's domain Kozachkov et al. (2025)). In our simulations, this same coupling produces spatially structured activity in the astrocyte processes ($p_{ij}^s$): processes near activity centers exhibit higher peak and more sustained activity than peripheral ones, reflecting their stronger integrated neighborly coupling.

This inherent encoding of spatial relationships within simulated STP dynamics provides a strong biological rationale for incorporating a similar distance-based relative positional information scheme in our Astromorphic Attention. We translate this observed principle into the Astrocyte-Modulated Hebbian Weight ($H_{astro}$) component via the term $\phi(R)$, using the positional encoding matrix $R$ (computed as described in the first paragraph). Calculating $H_{astro} = \frac{1}{m}\phi(R)^T V$ (Eq. 4) thus integrates a form of spatial context whose use is directly motivated by its analogy to simulated astrocyte STP behavior. This offers a flexible, learnable, and biologically-grounded method for

incorporating relative positional context, distinct from standard approaches lacking this neuro-glial justification. Having addressed this spatially-informed component of attention, we now turn to the temporal memory mechanisms essential for processing long sequences.

### 3.3 ASTROCYTE-INSPIRED MEMORY MECHANISM

To effectively model long-range dependencies across the segments processed by RMAAT (Sec 3.2.1), we require a mechanism that not only propagates context but does so efficiently, reflecting biological principles of memory consolidation. We draw inspiration from the Long-Term Plasticity (LTP) dynamics associated with astrocytes, particularly the behavior of the long-term astrocyte process parameter ($p_{ij}^l$ in Eq. 3), which integrates synaptic activity ($s_{ij}$) over extended timescales ($\tau_p^l$).

**Computational Macro Model (Contribution 1):** We leverage the detailed computational neuroscience model of neuron-astrocyte interactions (Sec 3.1) to understand the principles underlying LTP. Simulations of this detailed model reveal essential characteristics of the LTP-related state ($p_{ij}^l$): gradual integration of information over successive Short-Term Plasticity (STP) cycles, continuous accumulation across these cycles, and eventual saturation. Figure 3 illustrates this simulated behavior for a $3 \times 3$ neuron network over 300 seconds (encompassing six 50s STP cycles). We distill these observed characteristics—temporal integration and saturation—into a **computational macro model** that captures the emergent dynamics of LTP-based memory consolidation at an abstract level, independent of specific architectural choices.

**From Macro Model to ML Architecture (Contribution 2):** While the macro model captures the biological principle of saturating memory integration, it does not directly prescribe how to implement this within a recurrent transformer processing sequences segment-by-segment. Contribution 2 bridges this gap by deriving a **Memory Retention Factor** that instantiates the macro model's saturation curve as a concrete compression schedule for RMAAT's memory tokens in segment-based processing.

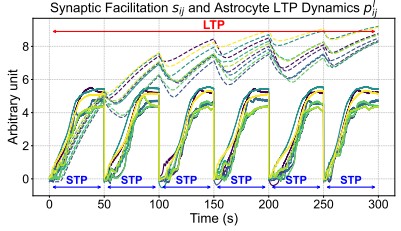

Figure 3: Simulation of the computational neuroscience model ($3 \times 3$-neuron network (9 connections), 300s total time, $6 \times 50$s STP cycles: STP cycles are reset every 50s in the 300s simulation) illustrating temporal integration for astrocyte-inspired memory. Dashed lines show the long-term astrocyte process ($p_{ij}^l$) integrating information and gradually saturating across STP cycles. Solid lines show the faster synaptic facilitation dynamics ($s_{ij}$) within each STP cycle.

To construct this factor, we normalize the macro model's total memory capacity (the integrated LTP signal at saturation) to 1 unit, then compute each segment's fractional contribution. Formally, for a sequence with $T$ segments, the Memory Retention Factor for segment $t$ is:

$$\text{RetentionFactor}(t, T) = \frac{\Delta p_t^l}{\sum_{i=1}^{T} \Delta p_i^l} \quad (6)$$

where $\Delta p_t^l$ represents the incremental increase in the simulated LTP state during segment $t$, obtained by running the LTP macro model (Eq. 3) for $T$ segments and measuring $\Delta p_t^l = p^l(t \cdot \tau_{cycle}) - p^l((t-1) \cdot \tau_{cycle})$, with $\tau_{cycle}$ being the duration of one STP cycle. Figure 4 illustrates the resulting factors for sequence lengths of 2, 4, 6, and 8 segments, showing that each subsequent segment contributes a diminishing fraction—enabling adaptive compression where the model adjusts retention based on anticipated sequence length, mimicking biological resource constraints.

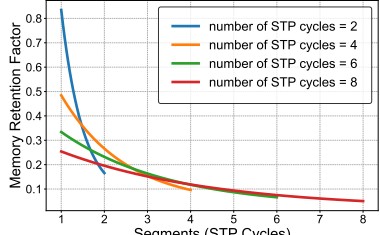

Figure 4: Memory Retention Factor derived from simulating the LTP macro model for different total sequence lengths (represented as total number of STP cycles from 2 to 8). The factor decreases per segment as the total sequence length increases, implementing adaptive, bio-inspired context compression.

**Application to Memory Tokens:** This Memory Retention Factor is applied directly to RMAAT's persistent **Memory Tokens** $mem_t \in \mathbb{R}^{M \times d}$ (Sec 3.2.1), which carry context across segments. As

tokens are updated within a segment via Astromorphic Attention to produce the next state $mem_{t+1}$, the factor corresponding to the current segment number and total sequence length (Figure 4) scales the updated state (e.g., $mem_{t+1} = \text{RetentionFactor}(t, \text{TotalSegments}) \times mem'_{t+1}$; see Algorithm 1 in Section 3.4 for the full pseudocode and Appendix D for additional implementation notes). This implements the adaptive compression dictated by the LTP macro model principle, ensuring memory remains bounded by gradually compressing older information. This contrasts with architectures like RMT (Bulatov et al., 2022) that often rely on fixed-size external memory slots updated via standard mechanisms, lacking this specific bio-inspired, adaptive compression rationale derived from LTP dynamics. This integrated, astrocyte-inspired memory system not only manages long-range context efficiently but also enables the resource-efficient AMRB training algorithm detailed next (Section 3.4).

## 3.4 AMRB Training Algorithm

Training recurrent architectures on long sequences via standard BPTT is often memory-prohibitive due to storing activations for the entire sequence length. To overcome this while leveraging RMAAT's unique memory structure, we introduce the Astrocytic Memory Replay Backpropagation (AMRB) algorithm (**Contribution** 3), an efficient training approach inspired by techniques for recurrent networks (Bellec et al., 2019; Meng et al., 2023)The core idea of AMRB is to avoid storing all intermediate activations within each segment during the forward pass. Instead, it leverages the persistent, compressed **Memory Tokens** ($mem_t$) described in Section 3.3. During the forward pass through $T_{seg}$ segments, only the sequence of memory token states ($mem_1, mem_2, ..., mem_{T_{seg}+1}$) passed between segments is stored in a replay buffer. During the backward pass, gradients are computed segment by segment. To calculate the gradients for segment $t$, the algorithm first retrieves the initial memory state $mem_t$ from the buffer. It then *recomputes* the forward pass for segment $t$ only, starting from $mem_t$ and using the input tokens $x_t$. This recomputation generates the necessary activations for calculating local gradients within segment $t$. The gradient flowing back from the subsequent segment $t + 1$ (which is initialized using the stored $mem_{t+1}$) is then backpropagated through the recomputed segment $t$, including the update path for $mem_t$. This process is repeated backward for all segments. Algorithm 1 below provides a detailed pseudocode description. The "replay" aspect refers to this recomputation of the forward pass for each segment during the backward pass, using the stored astrocyte-inspired memory state as the starting point.

---

**Algorithm 1:** Astrocytic Memory Replay Backpropagation (AMRB)

**Input:** $rollout = [x_1, x_2, ..., x_T]$: List of input tokens for $T$ time steps.
**Input:** $m_1$: Initial memory state (input to step $t = 1$).
**Output:** $m_{T+1}$: Updated memory state (output after step $t = T$).

1  Initialize $replay\_buffer \leftarrow []$ Append $m_1$ to $replay\_buffer$ ;                    // Store initial input state $m_1$
   /* Forward Pass                                                                                                          */
2  **for** $t = 1$ **to** $T$ **do**
3      $m'_{t+1} \leftarrow \text{Model}(x_t, m_t)$ ;                                   // Compute intermediate state (no grad)
4      $m_{t+1} \leftarrow \text{RetentionFactor}(t, TotalSegments) \times m'_{t+1}$ ;                      // Apply retention factor
5      **if** $t < T$ **then**
6         Append $m_{t+1}$ to $replay\_buffer$ ;     // Store input state $m_{t+1}$ for next step's recomputation

   /* Backward Pass                                                                                                         */
7  Initialize $\nabla m_{T+1} \leftarrow 0$ ;                                             // Init gradient for state after last step
8  **for** $t = T$ **to** $1$ **do**
9      Retrieve $m_t$ from $replay\_buffer$ ;                                      // Get input state for segment $t$
10     $m'_{t+1}, o_t \leftarrow \text{Model}(x_t, m_t)$ ;                              // Recompute segment $t$ (track grads)
11     Compute loss $L_t \leftarrow \text{loss\_function}(o_t)$ ;                       // Compute loss for current step
12     Perform backpropagation: $L_t.\text{backward}()$ ;                     // Compute param grads $\partial L_t/\partial \theta_t$, etc.
13     $m'_{t+1}.\text{backward}(\text{gradient} = \nabla m_{t+1}, \text{retain\_graph=True})$ ;             // Compute $\nabla m_t$ via chain rule

14 Save $m_{T+1}$ for the next rollout's update

---

AMRB offers significant memory efficiency and potential speed advantages. Unlike standard BPTT, which stores extensive activations, AMRB only caches the compact set of $M$ memory tokens passed between the $T_{seg}$ segments. Since $M$ is typically small, the memory footprint is drastically reduced. While AMRB involves recomputing activations during backpropagation, the associated memory saving often outweighs the recomputation cost for very long sequences, potentially leading to faster overall training (further details in Section 4) compared to standard BPTT.

In summary, RMAAT introduces a novel astrocyte-inspired adaptive memory compression system. This is realized through the Memory Retention Factor derived from simulated astrocyte LTP, provid-

ing a principled, non-learned method for compressing and propagating context between segments. Our core hypothesis is that this principled compression enables a more efficient training strategy. By structuring the flow of information into a compressed set of memory tokens, we can forgo backpropagation through every token—the source of high memory costs in standard BPTT used in RMT—and instead use our highly memory-efficient AMRB algorithm. Our ablation study in Section 4.2 validates this critical synergy: removing the compression causes a significant accuracy drop. This demonstrates that our bio-inspired compression is crucial for making the memory-saving AMRB algorithm effective, and this combination is directly responsible for RMAAT's gains in both efficiency and accuracy.

# 4 EXPERIMENTS

## 4.1 EXPERIMENTAL SETUP AND RESULTS

**Benchmark, Setup and Baselines:** We evaluate RMAAT using the Long Range Arena (LRA) benchmark (Tay et al., 2020). Models are implemented in PyTorch and trained from scratch (details in Appendix E). We evaluate RMAAT against the standard Transformer and a selection of prominent efficient Transformer models. While recent works have ventured into alternative frameworks like State-Space Models (SSMs) Gu et al. (2021); Gu & Dao (2023), we include these established efficient Transformers as they represent a direct lineage of architectural modifications for efficiency, providing a relevant context for RMAAT's approach which prioritizes deeper biological plausibility over purely mathematical or structural innovations. For a focused comparison of its recurrent and bio-inspired elements, we include key iso-architecture baselines. These are: *Astromorphic Transformer (AT)* (Mia et al., 2025), a non-recurrent model with astrocyte features but missing RMAAT's recurrence and memory; *Recurrent Memory Transformer (RMT)* (Bulatov et al., 2022), which processes segments recurrently with memory tokens but uses standard attention and lacks RMAAT's specific memory compression or training; and *Recurrent Linear Transformer (RLT)*, based on the *Linear Transformer (LT)* (Katharopoulos et al., 2020), implemented with recurrent structure and memory tokens as RMAAT but without RMAAT's specific memory retention factor, AMRB training, or the enhanced positional encoding and non-linearity found in (Mia et al., 2025). These latter models serve as important iso-architecture baselines to isolate the effects of RMAAT's contributions.

Table 1: Accuracy and Memory Comparison on Long Range Arena (LRA) Benchmark Tasks.

| Model | ListOps ($2K$) | | Text ($4K$) | | Retrieval ($8K$) | | Image ($1K$) | | Pathfinder ($1K$) | | Average |
|---|---|---|---|---|---|---|---|---|---|---|---|
| | Acc.(S.) | Mem.[*] | Acc.(S.) | Mem.[*] | Acc.(S.) | Mem.[*] | Acc.(S.) | Mem.[*] | Acc.(S.) | Mem.[*] | Acc. |
| Transformer (Vaswani et al., 2017) | 36.4(1) | 4.7 | 64.3(1) | 6.7 | 57.5(1) | 5.2 | 42.4(1) | 7.8 | 71.4(1) | 5.4 | 54.4 |
| Sparse Trans.[a](Child et al., 2019) | 17.1(1) | — | 63.6(1) | — | 59.6(1) | — | 44.2(1) | — | 71.7(1) | — | 51.2 |
| Longformer[a](Beltagy et al., 2020) | 35.6(1) | — | 62.9(1) | — | 56.9(1) | — | 42.2(1) | — | 69.7(1) | — | 53.5 |
| Linformer[a](Wang et al., 2020) | 35.7(1) | — | 53.9(1) | — | 52.3(1) | — | 38.6(1) | — | 76.3(1) | — | 51.4 |
| Reformer[a](Kitaev et al., 2020) | 37.3(1) | — | 56.1(1) | — | 53.4(1) | — | 38.1(1) | — | 68.5(1) | — | 50.7 |
| BigBird[a](Zaheer et al., 2020) | 36.1(1) | — | 64.0(1) | — | 59.3(1) | — | 40.8(1) | — | 74.9(1) | — | 55.0 |
| LT (Katharopoulos et al., 2020) | 16.1(1) | 4.7 | 65.9(1) | 5.7 | 53.1(1) | 3.9 | 42.3(1) | 6.2 | 75.3(1) | 6.2 | 50.5 |
| Performer[a](Choromanski et al., 2020) | 18.0(1) | — | 65.4(1) | — | 53.8(1) | — | 42.8(1) | — | 77.1(1) | — | 51.4 |
| FNet[c](Lee-Thorp et al., 2021) | 35.3(1) | — | 65.1(1) | — | 59.6(1) | — | 38.7(1) | — | 77.8(1) | — | 55.3 |
| Nyströmformer[c](Xiong et al., 2021) | 37.2(1) | — | 65.5(1) | — | 79.6(1) | — | 41.6(1) | — | 70.9(1) | — | 59.0 |
| Luna-256[c](Ma et al., 2021) | 37.3(1) | — | 64.6(1) | — | 79.3(1) | — | 47.4(1) | — | 77.7(1) | — | 61.3 |
| AT (Mia et al., 2025) | 18.1(1) | 4.7 | 61.5(1) | 5.8 | 77.3(1) | 4.1 | 47.3(1) | 6.2 | 77.9(1) | 6.3 | 56.4 |
| RMT (Bulatov et al., 2022) | 37.4(8)[b] | 20.4 | 65.0(8) | 24 | 79.3(16) | 18.3 | 54.6(2) | 22.7 | 81.5(4) | 12.7 | 63.6 |
| RLT (Kozachkov et al., 2023) | 18.4(8)[b] | 14.4 | 64.8(8) | 22.6 | 78.4(16) | 12.1 | 55.0(2) | 21.6 | 74.9(4) | 13.6 | 58.3 |
| **RMAAT (Ours)** | **38.9(8)[b]** | **5.2** | **65.9(8)** | **5.1** | **83.2(16)** | **3.4** | **64.8(2)** | **5.3** | **87.1(4)** | **4.7** | **68.0** |

[*] Acc.(S.): Accuracy(%) (Segments used). Mem. (GB): Peak GPU Memory.
[a] These models might have varying sequence lengths in a single segment compared to others. Results are referenced from (Tay et al., 2020).
[b] ListOps ($8K$) length used for segment calculation, resulting in 8 segments each with 1024 sequence length.
[c] These results are referenced from paper (Gu et al., 2021).

**Performance and Throughput Results:** Table 1 presents the main accuracy and memory usage results. It compares RMAAT against baselines across the five LRA tasks, showing accuracy percentages (and segments used for recurrent models) alongside peak GPU memory consumption in GB. RMAAT demonstrates competitive accuracy, particularly on longer context tasks like Retrieval, while maintaining significantly lower memory usage compared to iso-architecture recurrent baselines. Table 2 details the training throughput. For non-recurrent models (LT, AT), speed is measured relative to the standard Transformer baseline ($1\times$). For recurrent models (RLT, RMAAT), speed is measured relative to the iso-architecture RMT baseline ($1\times$) to better isolate the impact of the attention mechanism and training algorithm within a recurrent framework. RMAAT exhibits significantly faster training speeds compared to RMT, achieving up to $1.73\times$ speedup on the Retrieval task. This

highlights the efficiency gains from the AMRB training algorithm combined with the $O(N)$ complexity of the astromorphic attention framework, compared to RMT's standard BPTT and $O(N^2)$ attention. To validate the contributions of RMAAT's core components, we performed several ablation studies, primarily focusing on the long-context Byte-Level Document Retrieval ($8K$) task, supplemented by sensitivity analysis on other tasks (See Appendix F).

### 4.2 ABLATION STUDIES

**Memory Retention Factor (Contributions** 1 **&** 2**):** Removing the retention factor significantly reduced accuracy on the Retrieval task ($83.2\% \rightarrow 80.5\%$) without changing memory usage (3.4 GB), confirming its vital role in context compression derived from the LTP macro model. This effect is consistent across modalities: on the Text

Table 2: Detailed Throughput/Speed Comparison on Long Range Arena (LRA) Tasks.

| Model | ListOps | Text | Retrieval | Image | Pathfinder |
|---|---|---|---|---|---|
| Transformer (Vaswani et al., 2017) | $1\times$ | $1\times$ | $1\times$ | $1\times$ | $1\times$ |
| LT (Katharopoulos et al., 2020) | $1.24\times$ | $1.01\times$ | $1.03\times$ | $1.03\times$ | $1.03\times$ |
| AT (Mia et al., 2025) | $1.26\times$ | $1.26\times$ | $1.05\times$ | $1.08\times$ | $1.03\times$ |
| RMT (Bulatov et al., 2022) | $1\times$ | $1\times$ | $1\times$ | $1\times$ | $1\times$ |
| RLT (Kozachkov et al., 2023) | $1.05\times$ | $1.13\times$ | $1.37\times$ | $1.21\times$ | $0.95\times$ |
| **RMAAT (Ours)** | **$1.5\times$** | **$1.5\times$** | **$1.73\times$** | **$1.3\times$** | **$0.95\times$** |

($4K$) task, accuracy decreases from $65.9\%$ to $64.9\%$ when the retention factor is removed.

**AMRB Training (Contribution** 3**):** Replacing AMRB with standard BPTT yielded similar accuracy but drastically increased peak memory usage on both Retrieval (3.4 GB $\rightarrow$ 15.0 GB, $\sim 4.4\times$) and Text (5.1 GB $\rightarrow$ 22.0 GB, $\sim 4.3\times$), demonstrating AMRB's memory efficiency benefits.

**RLT + AMRB Ablation (Astromorphic Components):** We also evaluate an RLT + AMRB variant, which applies the Memory Retention Factor and AMRB to the Recurrent Linear Transformer (RLT) baseline (standard linearized attention, without $H_{astro}$ or $P$). On Retrieval ($8K$), RLT + AMRB attains $79.2\%$ accuracy with peak memory $\sim 3.4$ GB and RMAAT-like throughput, versus $78.4\%$ and 12.1 GB for RLT and $83.2\%$ and 3.4 GB for RMAAT.

**Applicability to Recurrent Architectures:** While the Memory Retention Factor and AMRB could potentially offer memory savings if applied to recurrent architectures like RMT (Bulatov et al., 2022), RMT's reliance on $O(N^2)$ softmax attention creates a forward pass bottleneck, limiting throughput gains. Furthermore, RMT would not benefit from the speed improvements associated with the non-linearity and relative positional encoding inherent in the astromorphic attention mechanism, as reported by Mia et al. (2025).

**Total Sequence Length:** Evaluating performance on shorter total sequence lengths by reducing the number of segments (while keeping segment sequence length constant at 512) resulted in significant accuracy drops in the Retrieval task (e.g., $71.5\%$ for 8 segments [$4K$ total length], $65.3\%$ for 4 segments [$2K$ total length] vs. $83.2\%$ for the baseline 16 segments [$8K$ total length]), demonstrating that the model benefits from processing the full context length allowed by the segmentation strategy.

**Other Hyperparameters:** Further analyses in Appendix F investigate sensitivity to other hyperparameters: spatial range (scale) of the positional encoding and number of memory tokens ($M$).

## 5 CONCLUSION

This work introduced the Recurrent Memory Augmented Astromorphic Transformer (RMAAT), demonstrating an effective approach to efficient long-sequence modeling by integrating computationally abstracted principles from astrocyte function. By incorporating astrocyte-inspired mechanisms for temporal memory compression and resource-efficient training (AMRB), RMAAT achieves strong average accuracy on the diverse Long Range Arena benchmark. This performance, coupled with competitive memory efficiency compared to standard and recurrent baselines (Table 1), validates the potential of leveraging neuro-glial principles for challenging sequence tasks within this benchmark setting. While promising, the current evaluation is primarily focused on LRA; future work should explore broader domains (including richer vision and multimodal settings), larger model scales, and extensions to true streaming or online processing where the total sequence length is not known in advance, as well as deeper theoretical analysis comparing RMAAT to related sequence model formalisms. Investigating additional neuro-glial computational mechanisms (such as astrocyte-astrocyte communication) and developing specialized hardware implementations also present exciting avenues. In conclusion, RMAAT highlights the value of neuroscience-algorithm co-design, suggesting that astromorphic computing is a promising direction for developing powerful and efficient AI systems capable of handling complex, long-range sequential data.

ACKNOWLEDGMENTS

Research was sponsored in part by the Army Research Office and was accomplished under Grant Number W911NF-24-1-0127. The views and conclusions contained in this document are those of the authors and should not be interpreted as representing the official policies, either expressed or implied, of the Army Research Office or the U.S. Government. The U.S. Government is authorized to reproduce and distribute reprints for Government purposes notwithstanding any copyright notation herein. This work was also supported partially by the National Science Foundation under award No. EFRI BRAID #2318101.

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

# A COMPUTATIONAL NEUROSCIENCE MODEL DETAILS

This appendix provides the detailed equations and parameters for the computational neuroscience model of the neuron-astrocyte network, which forms the foundation for the mechanisms abstracted in RMAAT, as discussed in Section 3.1. The model integrates dynamics across different timescales, capturing key aspects of neuronal activity, synaptic plasticity, and astrocytic modulation.

## A.1 NEURAL DYNAMICS

The membrane potential $V_i(t)$ of neuron $i$ is modeled using Leaky Integrate-and-Fire (LIF) dynamics. The evolution of the membrane potential is given by:

$$\tau_n \frac{dV_i(t)}{dt} = -\lambda(V_i(t) - V_{reset}) + I_i(t) \tag{7}$$

where:

- $\tau_n$: Neural membrane time constant ($R_m C_m$).
- $V_i(t)$: Membrane potential of neuron $i$ at time $t$.
- $\lambda$: Decay rate for the membrane potential.
- $V_{reset}$: Reset potential after a spike.
- $I_i(t)$: Total input current to neuron $i$.

When $V_i(t)$ reaches a threshold $V_{th}$, the neuron fires a spike, and $V_i(t)$ is reset to $V_{reset}$. The neuron's activity level, $x_i$, conceptually represents its firing rate or probability, influenced by $V_i$.

The input current $I_i(t)$ is determined by synaptic inputs modulated by synaptic facilitation $s_{ij}$ and an intrinsic bias $b_i$:

$$I_i(t) = \sum_{j=1}^{N} g(s_{ij}) S_j(t) + b_i \tag{8}$$

where:

- $g(s_{ij})$: Effective synaptic weight, dependent on synaptic facilitation $s_{ij}$. Typically a non-linear function, e.g., sigmoid or linear.
- $S_j(t)$: Spike train from presynaptic neuron $j$, often modeled as $\sum_k \delta(t - t_k^j)$ where $t_k^j$ are spike times.
- $b_i$: Intrinsic bias current for neuron $i$.
- $N$: Number of presynaptic neurons connected to neuron $i$.

## A.2 SYNAPTIC DYNAMICS

Synaptic facilitation $s_{ij}$ between postsynaptic neuron $i$ and presynaptic neuron $j$ captures short-term changes in synaptic efficacy. Its dynamics are influenced by neuronal co-activation and astrocyte modulation:

$$\tau_s \frac{ds_{ij}}{dt} = -\beta s_{ij} + \theta(x_i)\theta(x_j) + \psi(p_{ij}^s) + c_{ij} \tag{9}$$

where:

- $\tau_s$: Synaptic dynamics timescale.
- $s_{ij}$: Synaptic facilitation level between neurons $i$ and $j$.
- $\beta$: Decay rate of synaptic facilitation.
- $\theta(x)$: Non-linear function representing neuronal activity contribution (e.g., thresholding or sigmoid). $x_i, x_j$ are activity levels of neurons $i, j$.
- $\psi(p_{ij}^s)$: Contribution from the short-term astrocyte process $p_{ij}^s$ (modulation). $\psi$ is typically a non-linear function (e.g., sigmoid, tanh).
- $p_{ij}^s$: Short-term astrocyte process parameter associated with synapse $(i, j)$.
- $c_{ij}$: Baseline bias for synaptic facilitation.

## A.3 Short-Term Astrocytic Process Dynamics (STP)

The short-term astrocyte process parameter $p_{ij}^s$, conceptually related to local intracellular $Ca^{2+}$ dynamics near the synapse, evolves based on interactions with other astrocyte processes:

$$\tau_p^s \frac{dp_{ij}^s}{dt} = -\gamma^s p_{ij}^s + \sum_{k,l=1}^{N} T_{ijkl} \psi(p_{kl}^s) + d_{ij} \tag{10}$$

where:

- $\tau_p^s$: Timescale for short-term astrocyte dynamics.

- $p_{ij}^s$: Short-term astrocyte process state for synapse $(i, j)$.

- $\gamma^s$: Decay rate for $p_{ij}^s$.

- $T_{ijkl}$: Coupling tensor representing concentration fluxes or spatial influence between the astrocyte process associated with synapse $(i, j)$ and the process associated with synapse $(k, l)$. It depends on the relative spatial distance between these synapses. Specifically, $T_{ijkl} \propto \exp(-\text{distance}_{ij,kl} \times \text{scale})$. The term $\text{distance}_{ij,kl}$ refers to the Euclidean distance between the spatial midpoint of synapse $(i, j)$ and the spatial midpoint of synapse $(k, l)$.

- $\psi(p_{kl}^s)$: Non-linear function representing the influence of astrocyte process $p_{kl}^s$.

- $d_{ij}$: Baseline bias for the astrocyte process.

## A.4 Long-Term Astrocytic Process Dynamics (LTP)

The long-term astrocyte process parameter $p_{ij}^l$ integrates synaptic activity over longer timescales, contributing to persistent changes and memory:

$$\tau_p^l \frac{dp_{ij}^l}{dt} = -\gamma^l p_{ij}^l + \kappa(s_{ij}) \tag{11}$$

where:

- $\tau_p^l$: Timescale for long-term astrocyte dynamics ($\tau_p^l \gg \tau_p^s$).

- $p_{ij}^l$: Long-term astrocyte process state for synapse $(i, j)$.

- $\gamma^l$: Decay rate for $p_{ij}^l$.

- $\kappa(s_{ij})$: Non-linear function representing the influence of sustained synaptic facilitation $s_{ij}$ on the long-term process.

## A.5 Parameter Values for Simulation

The specific values used for the simulations presented in the main text (e.g., Figure 3) are listed in Table 3.

## B Astromorphic Attention Mechanism Details

This appendix provides an expanded description of the Astromorphic Attention mechanism employed within RMAAT segments (Section 3.2.2). This efficient mechanism, operating with $O(N)$ complexity, replaces the standard $O(N^2)$ self-attention. Its design is fundamentally inspired by computational models of the tripartite synapse, involving interactions between neurons and astrocytes (Appendix A), and specifically draws from principles of Short-Term Plasticity (STP) dynamics. We conceptualize the mechanism using a two-layer network structure (input/hidden and output layers) modulated by astrocyte-like computations (Figure 2). The process unfolds in two distinct operational phases: a Write Mode for context encoding and a Read Mode for context retrieval.

Table 3: Computational Neuroscience Model Hyperparameters Used in Simulations.

| Parameter | Value |
|---|---|
| *Network Details* | |
| Number of presynaptic neurons | 3 |
| Number of postsynaptic neurons | 3 |
| Number of astrocytes | 1 |
| Simulation Timescale | 300 s |
| STP Cycle Duration | 50 s |
| Timestep, $dt$ | 0.04 s |
| *Neural Dynamics* | |
| Neural dynamics timescale, $\tau_n$ | 0.5 s |
| Membrane potential threshold, $V_{th}$ | 1 mV |
| Reset potential, $V_{reset}$ | $-1$ mV |
| Decay parameter, $\lambda$ | 0.2 |
| Bias parameter, $b_i$ | 0 |
| Non-linearity, $\phi$ | $tanh$ |
| *Synaptic Dynamics* | |
| Synaptic dynamics timescale, $\tau_s$ | 0.75 s |
| Decay parameter, $\beta$ | 0.25 |
| Bias parameter, $c_{ij}$ | 0 |
| Non-linearity, $\theta$ | $tanh$ |
| *Astrocytic STP Dynamics* | |
| STP dynamics timescale, $\tau_p^s$ | 1 s |
| Decay parameter, $\gamma^s$ | 0.2 |
| Bias parameter, $d_{ij}$ | 0 |
| Non-linearity, $\psi$ | $tanh$ |
| *Astrocytic LTP Dynamics* | |
| LTP dynamics timescale, $\tau_p^l$ | 6 s |
| Decay parameter, $\gamma^l$ | 0.1 |
| Non-linearity, $\kappa$ | $sigmoid$ |

## B.1 NETWORK STRUCTURE AND INITIAL PROJECTIONS

The core computation is conceptualized within a network architecture comprising three functional layers: an input layer, a hidden layer, and an output layer. The input layer receives the segment's combined token representations $X \in \mathbb{R}^{N \times d}$, where $d$ is the embedding dimension. This input $X$ consists of $N_{seq}$ sequence tokens ($x_t$) concatenated with $M$ persistent memory tokens ($mem_t$), resulting in $N = N_{seq} + M$ total tokens per segment. The hidden layer consists of $m$ processing units (neurons), acting as an intermediate representation space. The output layer has $d$ units, matching the input embedding dimension, producing the final representation for the segment.

Initial processing involves linear projections of the input $X$ to generate the standard attention components: Keys ($K$), Queries ($Q$), and Values ($V$). These projections are facilitated by learnable weight matrices that map between the layers:

- $W_K \in \mathbb{R}^{d \times m}$ projects the $d$-dimensional input $X$ to the $m$-dimensional hidden space, producing Keys $K = X W_K \in \mathbb{R}^{N \times m}$. Keys represent the input signals as interpreted or encoded by the hidden layer units (presynaptic neurons).

- $W_Q \in \mathbb{R}^{d \times m}$ similarly projects $X$ to the hidden space, producing Queries $Q = X W_Q \in \mathbb{R}^{N \times m}$. Queries serve as the signals used later in the Read Mode to probe the encoded context.

- $W_V \in \mathbb{R}^{d \times d}$ projects $X$ directly to the output space dimension, producing Values $V = X W_V \in \mathbb{R}^{N \times d}$. Values represent the content or features associated with each input token relevant for constructing the output.

Following these projections, a non-linear activation function, $\phi$ (typically $\phi(x) = \text{elu}(x) + 1$), is applied element-wise to the Keys ($K$) and Queries ($Q$). The resulting $\phi(K)$ and $\phi(Q)$ represent the activated states of the hidden layer neurons, signifying their non-linear response to the key and query inputs, respectively. These activated states are central to the subsequent Write and Read mode computations.

## B.2 WRITE MODE: ENCODING CONTEXT

The Write Mode encodes contextual information from the entire segment by computing effective synaptic weights and an abstracted astrocyte state. Conceptually, this involves sequential updates as each token is processed, integrating Hebbian principles with astrocyte-inspired modulation. For efficient implementation, these sequential updates are typically realized through final matrix operations performed once per segment.

**Neuronal Hebbian Weight** ($H_{neuron}$)**:** This component represents the direct connection strength between the hidden (presynaptic) and output (postsynaptic) layers, learned via Hebbian plasticity.

- *Conceptual Per-Token Update:* As each token $t$ (from 1 to $N$) is processed, its activated key $h_t = \phi(k_t)$ (the $t$-th row of $\phi(K)$) and corresponding value $v_t$ (the $t$-th row of $V$) contribute to the weight update: $H_{neuron,t} = H_{neuron,t-1} + \frac{1}{m} h_t^T v_t$ (assuming $H_{neuron,0} = 0$).
- *Matrix Implementation:* The final weight after processing all $N$ tokens is efficiently calculated as the sum of these outer products:

$$H_{neuron} = \sum_{t=1}^{N} \frac{1}{m} h_t^T v_t = \frac{1}{m} \phi(K)^T V \quad \in \mathbb{R}^{m \times d} \tag{12}$$

This captures baseline Hebbian learning, linked to the $\theta(x_i)\theta(x_j)$ term (Eq. 9).

**Astrocyte-Modulated Hebbian Weight** ($H_{astro}$)**:** This component models the astrocyte's influence on the hidden-to-output connection, incorporating spatial context via a relative positional encoding matrix $R \in \mathbb{R}^{N \times m}$. The computation of $R$ itself, detailed in Section 3.2.3 and inspired by STP spatial dynamics ($T_{ijkl}$ in Eq. 10), involves transforming a base distance matrix ($r_{ij} = \exp(-\|\text{pos}_i - \text{pos}_j\| \times \text{scale})$) using learnable projections $M$ and $W_{rel}$ ($R = W_{rel}(MrM^T)$).

- *Conceptual Per-Token Update:* Similar to $H_{neuron}$, the astrocyte modulation associated with token $t$, represented by the $t$-th row of the activated positional encoding $\phi(R)$ (let us denote it as $\phi(r_t)$), updates the weight: $H_{astro,t} = H_{astro,t-1} + \frac{1}{m} \phi(r_t)^T v_t$ (assuming $H_{astro,0} = 0$).
- *Matrix Implementation:* The final weight is calculated across all tokens:

$$H_{astro} = \sum_{t=1}^{N} \frac{1}{m} \phi(r_t)^T v_t = \frac{1}{m} \phi(R)^T V \quad \in \mathbb{R}^{m \times d} \tag{13}$$

This functionally abstracts the astrocyte modulation term $\psi(p_{ij}^s)$ (Eq. 9).

**Presynaptic State** ($g$)**:** This vector abstracts the astrocyte's internal state (e.g., calcium level) responding to cumulative presynaptic activity from the hidden layer. The following two-stage view (linear accumulation of activated keys followed by a non-linear transformation on the total sum) aligns with how astrocytes might integrate signals over a period and then exhibit a saturated response.

- *Conceptual Per-Token Accumulation:* As each token $t$ (from 1 to $N$) is processed, its activated key $h_t = \phi(k_t)$ (the $t$-th row of $\phi(K)$) contributes to a running sum. If we denote this accumulating sum as $g_{acc}$, then $g_{acc,t} = g_{acc,t-1} + h_t$, starting with $g_{acc,0} = 0$. This represents the linear integration of presynaptic signals before the astrocyte's non-linear response.
- *Matrix Implementation and Incorporation of Astrocytic Non-linearity :* For the entire segment, the total accumulated influence from all $N$ tokens is first computed as the sum $\sum_{t=1}^{N} \phi(k_t)$. The non-linear saturation effect, modeled by the exponent $\alpha$, is then applied element-wise to this sum vector (which is of dimension $1 \times m$) to yield the final presynaptic state $g \in \mathbb{R}^{1 \times m}$ for the segment:

$$g = \left( \sum_{t=1}^{N} \phi(k_t) \right)^{\alpha} \tag{14}$$

This $g$ mirrors the temporal integration property of astrocyte processes ($p_{ij}^s$) in Appendix A.

**Combined Hebbian Weight** ($H$)**:** The total effective synaptic strength, $H \in \mathbb{R}^{m \times d}$, is the sum of the neuronal and astrocyte-modulated components:

$$H = H_{neuron} + H_{astro} \tag{15}$$

## B.3 READ MODE: RETRIEVING CONTEXT

The Read Mode utilizes the activated queries $\phi(Q)$ to retrieve the context encoded in the final aggregated weights ($H$) and state ($g$) computed during the Write Mode. This phase typically involves parallel matrix operations across all $N$ query tokens simultaneously.

**Interaction Strength / Calcium Response** ($C$)**:** Calculates the interaction ($C \in \mathbb{R}^{N \times 1}$) between the current active queries $\phi(Q)$ and the final presynaptic state $g$, representing the astrocyte's response.

$$C = \phi(Q)g^T \tag{16}$$

**Feedback Factor** ($P$)**:** Derives a feedback factor ($P \in \mathbb{R}^{N \times 1}$), usually inversely related to $C$, abstracting astrocyte feedback mechanisms.

$$P = 1/C \tag{17}$$

**Final Attention Output** ($L$)**:** Queries $\phi(Q)$ retrieve context from $H$, modulated element-wise ($\odot$) by the feedback $P$. A residual connection adds the original input $X$. The result $L \in \mathbb{R}^{N \times d}$ is the final output of the attention layer.

$$L = \phi(Q)(H \odot P) + X \tag{18}$$

The expanded form is:

$$L = \phi(Q) \left( \left( \frac{1}{m}(\phi(K)^T + \phi(R)^T)V \right) \odot \left( \frac{1}{C} \right) \right) + X \tag{19}$$

(where $C = \phi(Q) \left[ \left( \sum_{t=1}^{N} \phi(k_t) \right)^{\alpha} \right]^T$)

This formulation can be compared to standard linearized self-attention, often expressed as $SA(X) = \phi(Q)(\phi(K)^T V)$ normalized appropriately. As detailed previously, our equation for $L$ (before the residual connection) shares the core structure $\phi(Q)(\dots V)$, ensuring linear complexity. However, the astromorphic approach introduces two key modifications inspired by the tripartite synapse: (1) The aggregated context includes both direct neuronal correlations ($\phi(K)^T V$) and astrocyte-modulated spatial information ($\phi(R)^T V$) within $H$. (2) The retrieved context ($H$) is dynamically modulated element-wise by the feedback factor $P = 1/C$, which depends on the interaction between the current query $\phi(Q)$ and the aggregated presynaptic state $g$. This astrocyte-inspired modulation introduces a dynamic, context-dependent weighting absent in standard linear attention, while preserving the overall $O(N)$ complexity.

## B.4 COMPUTATIONAL COMPLEXITY

The Astromorphic Attention mechanism achieves $O(N)$ complexity per segment with respect to the sequence length $N$, assuming the hidden dimension $m$ and embedding dimension $d$ are constants relative to $N$. A detailed breakdown follows:

- **Write Mode Complexity Analysis:**
  - *Initial Projections (K, Q, V):* Calculating $K$, $Q$, and $V$ involves matrix multiplications ($XW_K$, $XW_Q$, $XW_V$) with complexities $O(Nmd)$, $O(Nmd)$, and $O(Nd^2)$ respectively. Activation $\phi$ adds $O(Nm)$.
  - *Hebbian Weights ($H_{neuron}, H_{astro}$):* Calculating $H_{neuron}$ ($\phi(K)^T V$) involves an $m \times N$ by $N \times d$ multiplication, costing $O(Nmd)$. Similarly, calculating $H_{astro}$ ($\phi(R)^T V$), assuming $R$ is computed efficiently, also costs $O(Nmd)$.
  - *Presynaptic State (g):* Summing $N$ vectors of size $m$ ($\sum \phi(k_t)$) costs $O(Nm)$. Applying the power $\alpha$ costs $O(m)$.

- *Combined Weight (H):* Addition costs $O(md)$.
- *Dominant Write Cost:* The most significant terms scale linearly with $N$, dominated by $O(Nmd)$ and $O(Nd^2)$. Crucially, the intermediate results $H$ and $g$ have dimensions independent of $N$.

- **Read Mode Complexity Analysis:**
  - *Interaction Strength (C):* Calculating $C$ ($\phi(Q)g^T$) is an $N \times m$ by $m \times 1$ matrix-vector multiplication, costing $O(Nm)$.
  - *Feedback Factor (P):* Calculating $P$ ($1/C$) is element-wise on an $N \times 1$ vector, costing $O(N)$.
  - *Final Output (L):* The main computation involves $\phi(Q)(H \odot P)$. The Hadamard product $H \odot P$ requires broadcasting $P$ and costs approximately $O(Nmd)$, if implemented by multiplying each row of $H$ by the corresponding element of $P$. The subsequent multiplication by $\phi(Q)$ ($N \times m$ by $m \times d$) costs $O(Nmd)$. The residual addition is $O(Nd)$.
  - *Dominant Read Cost:* The matrix multiplication dominates, scaling as $O(Nmd)$.

- **Overall Complexity and Comparison:**
  - Both Write and Read modes are dominated by operations scaling linearly with $N$ (primarily $O(Nmd)$). Therefore, the total complexity per segment is $O(N)$.
  - This linear scaling provides a significant advantage over standard self-attention, where the computation of the $N \times N$ attention score matrix ($QK^T$) leads to an overall complexity of $O(N^2d)$. The Astromorphic mechanism avoids this quadratic bottleneck by computing fixed-size intermediate representations ($H, g$) and using linear-time operations for context retrieval.

## C $\quad T_{ijkl}$ FORMATION AND VISUALIZATION

This appendix details the calculation and visualization of the spatial coupling tensor $T_{ijkl}$. This tensor is crucial in the Short-Term Astrocytic Process Dynamics (STP) described in Appendix A (Eq. 10), where it models the distance-dependent influence between different astrocyte processes associated with synapses $(i, j)$ and $(k, l)$. Understanding its structure helps motivate the bio-inspired relative positional encoding used in Section 3.2.3. The specific simulation results visualized in this appendix (e.g., $T_{ijkl}$ slices and $p_{ij}^s$ dynamics) were generated

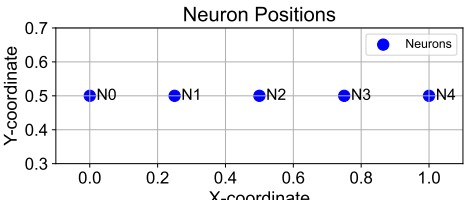

Figure 5: Spatial layout of the $N = 5$ neurons used in the simulation.

using a network size of $N = 5$ and a neural bias parameter $b = 0.1$, run for a duration of 50 seconds (representing one STP cycle), with all other model parameters set as detailed in Appendix A (Table 3).

### C.1 DISTANCE CALCULATION

For simulation purposes, we first define the spatial layout of the neurons. For the example shown, we consider $N = 5$ neurons arranged linearly in a 1D space, assigned coordinates for visualization (see Figure 5). A synapse $(i, j)$ connects post-synaptic neuron $i$ and presynaptic neuron $j$. We define the spatial position of synapse $(i, j)$ as the midpoint between the coordinates of neuron $i$ and neuron $j$. This results in a grid of $N \times N = 25$ possible synapse locations for $N = 5$ neurons (see Figure 6).

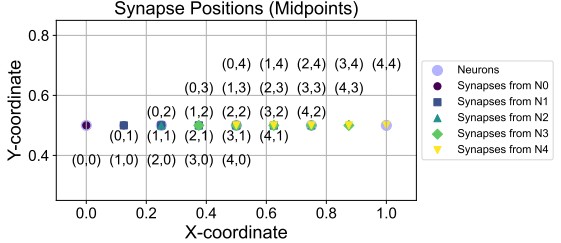

Figure 6: Calculated midpoint positions for all possible synapses between the $N = 5$ neurons.

## C.2 SYNAPSE POSITION CALCULATION

Next, we calculate the pairwise Euclidean distance, $\text{distance}_{ij,kl}$, between the midpoint coordinates of every pair of synapses $(i, j)$ and $(k, l)$. This forms a distance matrix capturing the spatial separation between all potential synaptic interaction sites (see Figure 7).

## C.3 $T_{ijkl}$ FORMULA AND VISUALIZATION

The coupling tensor $T_{ijkl}$ models the strength of influence (e.g., via concentration fluxes like calcium diffusion) between the astrocyte process at synapse $(i, j)$ and the process at synapse $(k, l)$. We model this influence using an exponential decay based on the calculated distance:

$$T_{ijkl} = \exp(-\text{distance}_{ij,kl} \times \text{scale}) \quad (20)$$

Here, scale is a positive parameter that controls the rate of spatial decay. A larger scale value leads to a faster decay, meaning interactions are more localized, while a smaller scale value allows for longer-range interactions.

Visualizing slices of the $T_{ijkl}$ tensor helps to understand the spatial interaction profile *from* a specific source synapse $(i, j)$ *to* all possible target synapses $(k, l)$. Figure 8 shows examples for source synapses located at the corner, edge, and center of the $5 \times 5$ grid, for different values of scale. Brighter colors indicate stronger influence (smaller distance or smaller scale). Notice how the spatial extent of the influence changes significantly with the scale parameter.

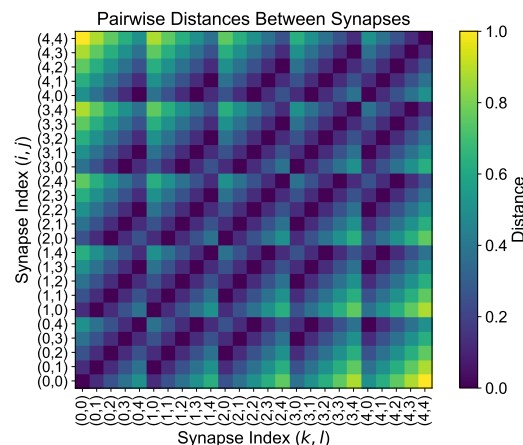

Figure 7: Heatmap visualizing the pairwise Euclidean distances between all synapse midpoints for $N = 5$, i.e., $\text{distance}_{ij,kl}$.

## C.4 IMPACT ON ASTROCYTE DYNAMICS AND LINK TO POSITIONAL ENCODING

The formulation of $T_{ijkl}$ as an exponential decay of distance (Eq. 20) implies that closer synaptic processes have a stronger potential for direct influence. However, the ultimate astrocytic response, represented by the short-term process dynamics $p_{ij}^s$ (Eq. 10), is not solely determined by $T_{ijkl}$. It results from the complex interplay of neuronal activity $(x_i, x_j)$, synaptic facilitation $(s_{ij})$, and the integrated influence from all other astrocyte processes $(\sum_{k,l} T_{ijkl} \psi(p_{kl}^s))$. Therefore, while $T_{ijkl}$ defines the strength of individual pairwise couplings, it is the simulation of the entire neuron-astrocyte network that reveals how these distance-dependent couplings translate into spatially modulated astrocytic responses over time.

As shown in Figure 9, these simulations demonstrate that synapses located centrally indeed tend to exhibit different temporal dynamics for $p_{ij}^s$ (e.g., higher peak and sustained activity) compared to those at corners or edges. This occurs because central locations benefit

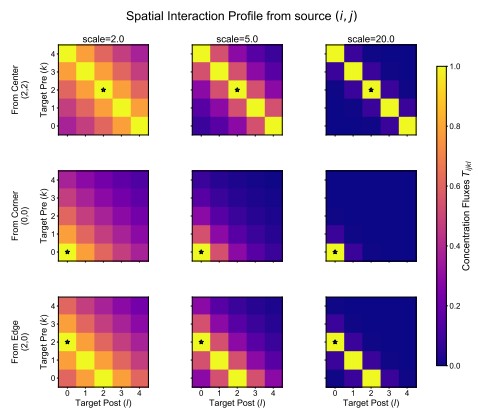

Figure 8: Visualization of $T_{ijkl}$ slices showing interaction strength from different source synapse locations (rows: center, corner, edge) to all target synapses $(k,l$ grid) for varying 'scale' parameters (columns: $2.0, 5.0, 20.0$).

from a stronger integrated influence from a larger number of relatively closer neighbors, as dic-

tated by the $T_{ijkl}$ coupling strengths. Changing the scale parameter in $T_{ijkl}$ further alters the range and strength of these interactions, consequently affecting the resulting $p_{ij}^s$ dynamics.

It is this simulated evidence—that the distance-dependent coupling encoded in $T_{ijkl}$, when integrated within the full system dynamics, leads to spatially modulated astrocytic STP responses ($p_{ij}^s$)—that provides the biological motivation for incorporating relative positional information in the astromorphic attention mechanism. Specifically, the base relative positional matrix $r$, defined in Section 3.2.3, uses the same exponential decay $\exp(-\text{distance} \times \text{scale})$ allowing the model to learn a suitable spatial interaction range for encoding positional context.

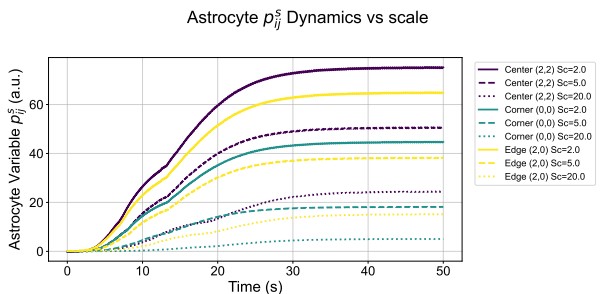

Figure 9: Simulated temporal dynamics of the short-term astrocyte process parameter $p_{ij}^s$ for synapses at different locations (center, corner, edge) under different spatial coupling 'scale' values (2.0, 5.0, 20.0). Simulation uses $N = 5$, $b = 0.1$.

## D  AMRB ALGORITHM

This appendix provides additional explanatory notes on the Astrocytic Memory Replay Backpropagation (AMRB) algorithm presented in Section 3.4 (Algorithm 1), focusing on gradient flow and implementation details.

**Notes:**

- The indexing convention in the algorithm follows $t = 1...T$, where $m_t$ is the input memory state to segment $t$, and $m_{t+1}$ is the output state.

- Model$(x_t, m_t)$ represents the forward pass computation for segment $t$, producing an intermediate memory state $m'_{t+1}$ and a segment output $o_t$.

- **Explanation of Line 13**: Line 13, '$m'_{t+1}$.backward(gradient $= \nabla m_{t+1}$, retain_graph=True)', is responsible for backpropagating the gradient from the subsequent segments' losses through the memory pathway of the current segment $t$.

  - **Context**: During the backward pass, for each segment $t$ (from $T$ down to 1), we first compute gradients arising from the local loss $L_t$ of that segment (Line 12). This step, $L_t$.backward(), calculates $\frac{\partial L_t}{\partial \theta_t}$ (gradients for model parameters $\theta_t$ in segment $t$) and also $\frac{\partial L_t}{\partial m_t}$ (gradient of local loss w.r.t. the input memory $m_t$).

  - $\nabla m_{t+1}$ **(Upstream Gradient)**: This is the gradient of the total loss from all future segments (i.e., segments $t + 1$ through $T$) with respect to $m_{t+1}$. The term $m_{t+1}$ is the memory state that segment $t$ passes to segment $t + 1$, calculated as $m_{t+1} =$ RetentionFactor $\times m'_{t+1}$. For the first iteration of this loop (when $t = T$), $\nabla m_{T+1}$ is typically initialized to zero as the final memory state does not directly contribute to a subsequent loss term.

  - **Operation of Line 13**: The command '$m'_{t+1}$.backward(gradient $= \nabla m_{t+1}$)' applies the chain rule. It takes the gradient $\nabla m_{t+1}$ (which is $\frac{\partial \text{Loss}_{\text{future}}}{\partial m_{t+1}}$, where $\text{Loss}_{\text{future}} = L_{t+1} + L_{t+2} + ... + L_T$) and computes the gradients of $\text{Loss}_{\text{future}}$ with respect to the inputs that formed $m'_{t+1}$. Specifically, it computes:

    * $\frac{\partial \text{Loss}_{\text{future}}}{\partial m_t}$ by backpropagating $\nabla m_{t+1}$ through the operations $m_{t+1} =$ RetentionFactor $\times m'_{t+1}$ and $m'_{t+1} = \text{Model}(x_t, m_t)$.

  * Additional contributions to $\frac{\partial \text{Loss}_{\text{future}}}{\partial \theta_t}$ by backpropagating through $\text{Model}(x_t, m_t)$. The automatic differentiation system handles the scaling by 'RetentionFactor' implicitly when applying the chain rule from $m_{t+1}$ to $m'_{t+1}$.

– **Accumulation of Gradients**: The gradients with respect to model parameters $\theta_t$ and input memory $m_t$ are accumulated. The gradient $\nabla m_t$ (which will be passed to segment $t-1$ in the next timestep) becomes the sum of the gradient from the local loss ($\frac{\partial L_t}{\partial m_t}$ from Line 12) and the gradient from future losses ($\frac{\partial \text{Loss}_{\text{future}}}{\partial m_t}$ computed in Line 13). Similarly, parameter gradients $\nabla \theta_t$ are also accumulated from both backpropagation steps.

– $retain\_graph = True$: The computational graph for segment $t$ (recomputed in Line 11) is used for two separate backward calls: one for $L_t$ (Line 12) and one for $m'_{t+1}$ (Line 13). $retain\_graph = True$ is necessary for the second call because the first call would typically free the graph. This ensures that intermediate activations and graph structure are available for both gradient computations within the current segment $t$.

# E  IMPLEMENTATION AND EXPERIMENTAL DETAILS

This appendix provides supplementary details regarding the experimental setup, hyperparameters, hardware/software environment, and measurement methodologies used for the experiments reported in Section 4.

## E.1  HYPERPARAMETERS

Key hyperparameters for RMAAT across the evaluated LRA tasks are summarized in Table 4. Consistent settings were used for iso-architecture baselines where applicable, with task-specific adjustments primarily for sequence length handling (Number of Segments) and training schedule (Epochs, Learning Rate). All models were trained from scratch using the AdamW optimizer and CrossEntropyLoss where applicable.

Table 4: Key Hyperparameters for RMAAT on LRA Tasks.

| Hyperparameters | ListOps (8K) | Text (4K) | Retrieval (8K) | Image (1K) | Pathfinder (1K) |
|---|---|---|---|---|---|
| **Training Parameters** | | | | | |
| Batch Size | 128 | 64 | 16 | 24 | 128 |
| Max Seg Len ($N$) | 1024 | 512 | 512 | 512 | 256 |
| Epochs | 50 | 100 | 50 | 50 | 100 |
| Learning Rate | $5.0e^{-4}$ | $1.5e^{-5}$ | $5.0e^{-5}$ | $5.0e^{-4}$ | $3.0e^{-5}$ |
| **Model Architecture** | | | | | |
| Embedding Dim ($d$) | 256 | 784 | 512 | 784 | 1024 |
| Number of Heads | 2 | 6 | 8 | 6 | 8 |
| FFN Dim | 1024 | 2048 | 2048 | 2048 | 2048 |
| Number of Encoder Layers | 1 | 1 | 1 | 3 | 1 |
| Dropout | 0.1 | 0.1 | 0.1 | 0.1 | 0.1 |
| **AMRB / Recurrence Parameters** | | | | | |
| Number of Segments ($T_{seg}$) | 8 | 8 | 16 | 2 | 4 |
| Number of Memory Tokens ($M$) | 8 | 32 | 4 | 32 | 4 |
| **Astromorphic Attention Parameters** | | | | | |
| Hidden Layer Neuron ($m$) | 100 | 100 | 100 | 100 | 100 |
| Non-linearity ($\alpha$) | 0.25 | 0.25 | 0.25 | 0.25 | 0.25 |
| **Positional Encoding Parameters** | | | | | |
| Rate of spatial decay (scale) | 2.0 | 2.0 | 2.0 | 2.0 | 2.0 |

## E.2  HARDWARE AND SOFTWARE

The experiments were conducted on a server with the following specifications:

- **OS:** Ubuntu 22.04.5 LTS (Kernel: Linux 6.8.0-52-generic x86_64)
- **CPU:** Intel(R) Xeon(R) Gold 6326 CPU @ 2.90GHz
- **RAM:** 503 GiB (approx. 512 GB)

- **GPU:** NVIDIA RTX A5000 (24GB Memory)
- **Software:** Models were implemented using PyTorch version 1.13.1 with CUDA version 11.7. Python version 3.10.13 was used.

### E.3 EFFICIENCY MEASUREMENT DETAILS

- **Peak GPU Memory:** Measured during the training process using standard GPU monitoring tools (e.g., `nvidia-smi` or PyTorch's memory management utilities) to capture the maximum memory allocated on the GPU.
- **Throughput/Speed:** Measured in terms of training time per epoch or overall training time, typically reported relative to a baseline (e.g., standard Transformer or RMT). Detailed results are in Table 2 (See Section 4).

## F ADDITIONAL RESULTS

This section provides additional details and sensitivity analyses complementing the main component ablation results summarized in Section 4.2. The above tables explore the sensitivity of RMAAT's performance (Accuracy) to variations in the positional encoding spatial range parameter (scale) and the number of memory tokens ($M$) on the ListOps ($8K$) and Text ($4K$) tasks. These sensitivity results illustrate the findings mentioned in Section 4.2: performance generally degrades when deviating significantly from the optimal values for scale and $M$, confirming the importance of tuning these hyperparameters for each task.

Table 5: RMAAT Accuracy (%) Sensitivity on ListOps ($8K$).

| Parameter | Value | Accuracy (%) |
|---|---|---|
| scale | 1.0 | 38.5 |
| | 2.0 | 38.9 |
| | 5.0 | 36.8 |
| | 10.0 | 36.5 |
| $M$ (Mem Tokens) | 2 | 37.9 |
| | 4 | 37.9 |
| | 8 | 38.9 |
| | 16 | 38.6 |
| | 32 | 38.4 |

Table 6: RMAAT Accuracy (%) Sensitivity on Text ($4K$).

| Parameter | Value | Accuracy (%) |
|---|---|---|
| scale | 1.0 | 65.4 |
| | 2.0 | 65.9 |
| | 5.0 | 65.1 |
| | 10.0 | 65.2 |
| $M$ (Mem Tokens) | 4 | 64.6 |
| | 16 | 65.4 |
| | 32 | 65.9 |
| | 64 | 65.0 |
| | 128 | 65.2 |

