# OpenReview forum: "RMAAT: Astrocyte-Inspired Memory Compression and Replay for Efficient Long-Context Transformers"
_ICLR.cc/2026/Conference — ICLR 2026 Poster_

### Official Review · Reviewer_RXES · 2025-10-27

**Soundness:** 2
**Presentation:** 2
**Contribution:** 3
**Rating:** 4
**Confidence:** 4

**Summary:**

The paper introduces a biologically inspired Transformer architecture that models neuron–astrocyte interactions to achieve efficient long-context processing. The proposed RMAAT integrates short-term and long-term plasticity (STP/LTP) into attention, possibly allowing the model to adaptively control how much information is remembered or forgotten over time. Through Astromorphic Attention, attention weights are dynamically adjusted based on astrocyte-like activity traces, while a Memory Retention Factor compresses and consolidates past context. The proposed AMRB method further reduces training memory by replaying compressed states. Experiments on the Long Range Arena benchmark show that RMAAT achieves good accuracy with low memory.

**Strengths:**

1. The paper introduces a novel architecture for long-context modeling by drawing from the biological mechanisms of astrocytes, both short-term and long-term plasticity are incoprated into modern attention-based architecture. It is well-motivated and the links to previous biological models are carefully explained.

2. The AMRB training algorithm reduces the memory burden during model training. The model achieves impressive performance on long-range-arena while maintaining low memory cost.

**Weaknesses:**

1. While this could be subjective, I find the lengthy method section hard to read. It might be better to directly contrast the vanilla attention with the proposed astrocyte-inspired attention (as the authors did in the appendix), point out which parts correspond to long-term or short-term astrocytic mechanisms, and then explain the link to computational neuroscience models. I also find Figure 2 difficult to interpret, as there are too many interacting components in a small space. For the AMRB algorithm, the pseudo-code in the appendix is much clearer than the textual description in the main section.

2. The model is only evaluated on the Long Range Arena benchmark, which is somewhat limited. For example, including results on language modeling benchmarks (e.g., [1]) could strengthen the paper.

3. The proposed method introduces several interacting mechanisms (Astrocyte-Modulated Hebbian Weight, Feedback Factor calculated with Presynaptic State, Memory Retention Factor), but the effects of these components are not fully disentangled in the ablation study.

[1] The fineweb datasets: Decanting the web for the finest text data at scale, 2024

**Questions:**

1. For the Memory Retention Factor, it seems that the factor is determined based on the total length of the context, but how is this known *a priori*? This appears somewhat strange, especially given that the model is inspired by memory retention in animals, which face a continuous stream of input and must adaptively decide how much information to retain.

2. Hebbian plasticity is described as short-term plasticity in this work, which is somewhat confusing since it is not necessarily considered short-term in a neuroscience context. Is the Hebbian plasticity effectively “cut off” between segments? If the processing time is already linear in the number of tokens, why is it still necessary to divide the context into separate segments? It would also be helpful if the authors could briefly discuss the connection to prior work on neuromodulated Hebbian learning in RNNs (e.g., [1][2]).

3. I noticed that only memory consumption is reported when comparing models. Given the substantially modified attention architecture, I suspect that the forward and backward computation times may also differ significantly from the baselines. It would therefore be helpful to include those comparisons as well.

4. As mentioned in Weakness 3, could the authors expand the ablation study to isolate the contributions of different mechanisms? It seems that only the Retrieval task was used for the experiment removing the Memory Retention Factor; if time permits, it would be valuable to include ablation results across other Long Range Arena tasks.

5. The paper claims the state-of-the-art average accuracy on Long Range Arena benchmark (e.g. line 477), but isn't modern state-space models able to achieve better performance (e.g., [3])?

[1] Backpropamine: Training self-modifying neural networks with differentiable neuromodulated plasticity  [2] Hebbian and Gradient-based Plasticity Enables Robust Memory and Rapid Learning in RNNs
[3] Simplified State Space Layers for Sequence Modeling

---

> ### Author Response · Authors · 2025-11-19
> **Comment 1**
>
> We thank the reviewer for their thoughtful and detailed feedback. We address each point below and have implemented revisions that we believe substantially clarify and strengthen the manuscript.
> ### On Improving the Presentation of the Methods Section and Figure 2
> We appreciate the reviewer’s feedback on the presentation. We acknowledge that moving some key elements from the appendix to the main text would significantly improve readability. In our revision:
> -   We have **moved the AMRB pseudocode from the appendix into Section 3.4**.-   The reviewer is correct that a direct comparison of Astromorphic Attention and vanilla/linearized attention is helpful. However, the detailed, side-by-side comparison is the central contribution of cited prior work (Mia et al., 2025). To avoid redundancy, we have clarified this link in the text. For the reviewer's convenience, the key differences are: 1) Astromorphic Attention incorporates **non-linear calcium dynamics** that model saturation, unlike the linear updates in vanilla attention, and 2) it integrates a **bio-inspired relative positional encoding** derived directly from astrocytic spatial activity, rather than using a separate, additive mechanism. We ensure our revised manuscript more clearly references the prior work for the full detailed comparison.
> -   We have **reorganized Section 3** to make the model flow more readable: clarifying the pipeline from neuron–astrocyte simulations to a macro-model and then to concrete STP-like (within-segment attention) and LTP-like (cross-segment memory) mechanisms, and we now present the **explicit mathematical formula for the Memory Retention Factor (Eq. 6)** in Section 3.3.  Finally, we have **enlarged Figure 2** and **substantially expanded its caption** to give a clearer, step-by-step walkthrough of the components and data flow, addressing the concern that the original figure and methods description were difficult to follow.
> ### On the Evaluation Scope and Comparisons to State-of-the-Art
> We thank the reviewer for this important question, as it allows us to precisely situate our work within the broader NeuroAI landscape. Our primary goal is not to compete with the latest, highly-engineered architectures like Mamba, but to contribute to the **fundamental science of brain-inspired computing** by demonstrating a significant advancement in the distinct and nascent subfield of astromorphic models. The unique benefit of our approach is that it validates this novel, neuroscience-grounded paradigm by showing it can be scaled to a challenging long-context benchmark with competitive performance and superior memory efficiency.
>
> Perhaps it is helpful to view our contribution within the clear research trajectory of this specific field:
> 1.  **Foundational Theory:** The initial works (e.g., Kozachkov et al., 2023; 2025) were highly theoretical, establishing the mathematical principles of neuron-astrocyte computation. For instance, the recent PNAS paper (Kozachkov et al., 2025) focused entirely on proving memory capacity scaling laws using abstract simulations, **without training or evaluating on standard ML benchmarks.** While Kozachkov et al. (2023) validated their theoretical framework by extracting weights from pre-trained models (ALBERT-base, Vision Transformer), they similarly did not perform training or benchmark evaluation on standard ML tasks.
> 2.  **Initial Validation:** A subsequent key study (Mia et al., 2025) represented the next step, successfully applying these theories for the first time to **standard but smaller-scale ML datasets** like IMDB, CIFAR-10, and WikiText-2, demonstrating initial promise.
> 3.  **Scaling to Long-Context:** The present work, **RMAAT**, takes the critical next step. We scale these validated principles to a full, challenging benchmark specifically designed for long-range dependencies (LRA). By doing so, we show that this bio-inspired approach is not just a theoretical curiosity but a viable path toward practical, efficient, and performant models for long sequences.
>
> Therefore, our primary comparison point is the prior work on astrocytes, not the SOTA in general sequence modeling. We now explicitly highlight large-scale language modeling and richer vision/multimodal benchmarks as key next steps in the conclusion of the revised manuscript, as these domains will further test the generality and scalability of astromorphic principles beyond the long-context regime we focus on here.

---

> > ### Author Response · Authors · 2025-11-19
> > **Comment 2**
> >
> > ### On Disentangling Components in the Ablation Study
> > We thank the reviewer for this excellent suggestion. Our ablation study was designed to systematically validate the **novel contributions of the present work**: the **Memory Retention Factor** and the **AMRB training algorithm**. The other mechanisms mentioned by the reviewer, such as the Astrocyte-Modulated Hebbian Weight and the Feedback Factor, are indeed integral components of the foundational Astromorphic Attention block, which was established and ablated in a cited prior work **(Mia et al., 2025)**.
> >
> > To test our contributions, we focused on the 8K Document Retrieval task as it is the most demanding long-context task in the LRA benchmark. However, we agree that demonstrating the robustness of these contributions across different modalities is important. To this end, we have also run the key ablation study (removing the Memory Retention Factor) on the **4K Text classification task**. The results confirm our findings: performance on the Text task also drops, from 65.9% to 64.9%, at essentially unchanged memory. In addition, we report the complementary ablation where we keep the retention factor but replace AMRB with standard BPTT on the Text task, which leaves accuracy essentially unchanged but increases peak memory from 5.1 GB to 22.0 GB (approximately $\sim4.3\times$). In the revised manuscript, we have added these additional results to our ablation study in Section 4.2 to demonstrate the robustness of our contribution.
> > ### On the Memory Retention Factor and Pre-Known Sequence Lengths
> > This is an excellent question that touches on the limitations of many current long-context models. Our current implementation, like most models evaluated on benchmark datasets (including LRA), assumes a known, finite sequence length—a common approach in models with pre-determined context windows (e.g., Dai et al., 2019; Bulatov et al., 2022). We agree that handling a continuous, unbounded stream of input is the key challenge for building truly adaptive memory systems. This is an active area of research, with recent architectural proposals like StreamingLLM (Xiao et al., 2024) and Ring Attention (Liu et al., 2024) specifically designed to address it. We explicitly discuss this limitation and highlight it as a critical avenue for future research in the revised manuscript.
> > ### On the Conceptual Framing of Hebbian Plasticity as STP and the Necessity of Segmentation
> > -   **Hebbian Plasticity as STP:** We use the term "short-term plasticity" to describe the Hebbian updates because they operate *within* a given segment and are effectively re-initialized in the next. This is in contrast to the persistent memory tokens that carry information across segments, which represents the "long-term" aspect of our model's memory. This distinction between fast, segment-local plasticity and slower, persistent memory mechanisms aligns with established neuroscience literature on short-term vs. long-term plasticity (e.g., Debanne et al., 2023; Bicknell & Latham, 2024), where short-term plasticity operates on millisecond-to-second timescales within local circuits, while long-term changes persist across longer periods. In the revised manuscript, we make this conceptual framing explicit in Section 3.1 (`Foundational Computational Neuroscience Model`), where we explain how the STP and LTP components of the neuron–astrocyte model map onto fast, within-segment attention modulation and slower, cross-segment memory integration, citing these works.
> > -   **Necessity of Segmentation:** This is a crucial point. While the Astromorphic Attention mechanism has linear time complexity O(N), training a model on a very long, unsegmented sequence still incurs a massive memory cost from storing the entire computational graph for backpropagation. **Segmentation is the key to memory efficiency.** By breaking the sequence into manageable chunks, and using our principled compression and AMRB algorithm, we can process sequences of arbitrary length while keeping the peak memory footprint low and constant.

---

> > > ### Author Response · Authors · 2025-11-19
> > > **Comment 3**
> > >
> > > -   **Prior Work:** Thank you for pointing out the relevant papers on neuromodulated Hebbian learning, including Miconi et al. (2020) on "Backpropamine" and Duan et al. (2023) on "Hebbian and Gradient-based Plasticity Enables Robust Memory and Rapid Learning in RNNs." These works explore how neuromodulatory signals can gate or modulate Hebbian plasticity in RNNs, which shares conceptual similarities with our astrocyte-mediated modulation of synaptic plasticity. However, our approach differs in that we derive the modulation dynamics directly from computational neuroscience models of astrocyte-neuron interactions (specifically, the tripartite synapse model), rather than using learned neuromodulatory signals. Related work on theoretical models of dual-timescale plasticity (e.g., Bicknell & Latham, 2024) and astrocyte regulation of plasticity (e.g., Stasenko et al., 2023) also provides theoretical foundations for our approach. In the revised manuscript, we have added a concise related-work discussion in Section 2 (`Related Works and Main Contributions`) connecting these lines of work to our astrocyte-based formulation and have included the corresponding citations.
> > > ### On Reporting Throughput and Computation Time
> > > We thank the reviewer for this question. These results are included in the main paper. Specifically, **Table 2** reports the training throughput (sequences per second), which directly reflects the forward and backward computation times. As the reviewer suspected, our architecture is indeed significantly faster than the iso-architecture RMT baseline. For instance, on the challenging Retrieval task, RMAAT demonstrates a **1.73x higher throughput**, confirming its computational efficiency.
> > > ### On Clarifying Claims of State-of-the-Art Performance
> > > Thank you for pointing this out. We did not intend to claim state-of-the-art performance over all models, but rather to highlight our model's strong performance within the established LRA benchmark. In the revised manuscript, we have carefully reviewed the wording and now describe our results as **strong average accuracy on the LRA benchmark with competitive memory efficiency among the baselines we evaluate**, avoiding any unqualified state-of-the-art claim.
> > >
> > > **References:**
> > > - Mia, M. Z. A., Bal, M., & Sengupta, A. (2025). *Delving Deeper Into Astromorphic Transformers*. IEEE Transactions on Cognitive and Developmental Systems.
> > > - Kozachkov, L., Kastanenka, K. V., & Krotov, D. (2023). Building transformers from neurons and astrocytes. Proceedings of the National Academy of Sciences, 120(34), e2219150120.
> > > - Kozachkov, L., Slotine, J. J., & Krotov, D. "Neuron–astrocyte associative memory." *Proceedings of the National Academy of Sciences* 122.21 (2025): e2417788122.
> > > -   Xiao, G., et al. (2024). *Efficient Streaming Language Models with Attention Sinks*. In International Conference on Learning Representations (ICLR).
> > > - Liu, H., et al. (2024). *Ring Attention with Blockwise Transformers for Near-Infinite Context*. arXiv preprint arXiv:2310.01889.
> > > - Bulatov, A., Kuratov, Y., & Burtsev, M. (2022). *Recurrent memory transformer*. ArXiv, abs/2207.06881.
> > > - Beltagy, I., Peters, M. E., & Cohan, A. (2020). *Longformer: The long-document transformer*. arXiv preprint arXiv:2004.05150.
> > > - Rae, J. W., Potapenko, A., Jayakumar, S. M., & Lillicrap, T. P. (2019). *Compressive transformers for long-range sequence modelling*. arXiv preprint arXiv:1911.05507.
> > > - Dai, Z., Yang, Z., Yang, Y., Carbonell, J. G., Le, Q., & Salakhutdinov, R. (2019, July). Transformer-xl: Attentive language models beyond a fixed-length context. In Proceedings of the 57th annual meeting of the association for computational linguistics (pp. 2978-2988).
> > > - Miconi, T., et al. (2020). *Backpropamine: Training self-modifying neural networks with differentiable neuromodulated plasticity*. In Advances in Neural Information Processing Systems (NeurIPS).
> > > - Duan, Y., Jia, Z., Li, Q., Zhong, Y., & Ma, K. (2023). Hebbian and gradient-based plasticity enables
> > > robust memory and rapid learning in RNNs. arXiv preprint arXiv:2302.03235.
> > > - Debanne, D., & Inglebert, Y. (2023). *Spike timing-dependent plasticity and memory*. Current Opinion in Neurobiology, 80, 102707.
> > > - Bicknell, B. A., & Latham, P. E. (2024). *Fast and slow synaptic plasticity enables concurrent control and learning*. bioRxiv preprint.
> > > - Stasenko, S. V., & Kazantsev, V. B. (2023, September). Model of Astrocyte Regulation of
> > > Spike-Timing-Dependent Plasticity. In 2023 Fifth International Conference Neurotechnologies and
> > > Neurointerfaces (CNN) (pp. 86-89). IEEE.

---

> > > > ### Comment · Reviewer_RXES · 2025-11-24
> > > >
> > > > Thanks for the response. I agree that the revisions help improve the clarity of the paper.
> > > >
> > > > Regarding the ablations, the authors note that “The other mechanisms mentioned by the reviewer, such as the Astrocyte-Modulated Hebbian Weight and the Feedback Factor, are integral components of the foundational Astromorphic Attention block, which was established and ablated in a cited prior work (Mia et al., 2025).” However, since Mia et al. (2025) evaluates different tasks and does not include the Memory Retention mechanism, I believe it is still important to ablate these components in the present work in order to isolate their contributions in the long-range setting.
> > > >
> > > > A more critical concern—and what makes me hesitant to raise my score—is that the motivation for the current work remains unclear. On the one hand, the evaluation is limited, the model cannot realistically compete with state-of-the-art architectures, and the authors argue that “the primary goal is not to compete with the latest highly engineered architectures like Mamba, but to contribute to the fundamental science of brain-inspired computing.” This is understandable. But on the other hand, the main contribution of the paper appears to be engineering the model from Mia et al. (2025) to handle long-range contexts under limited GPU memory (as we don't really need the segmentation and memory retention factor if not for memory efficiency), and I do not see a particularly strong or novel contribution to the “fundamental science” the authors emphasize. This mismatch between the stated goals and the actual contributions weakens the overall positioning of the paper.

---

> > > > > ### Author Response · Authors · 2025-11-25
> > > > > **Comment 1**
> > > > >
> > > > > We thank the reviewer for their continued engagement and for raising these critical points. We appreciate the opportunity to further clarify our contributions and strengthen the evidence.
> > > > >
> > > > > ### 1. Additional Ablation: Disentangling Astromorphic Components
> > > > > We completely agree with the reviewer that isolating the components of the Astromorphic Attention block in the long-range setting is valuable. To address this, we have now performed the specific ablation the reviewer suggested implicitly: **RLT + AMRB**.
> > > > >
> > > > > The **Recurrent Linear Transformer (RLT)** baseline essentially represents RMAAT's architecture but with a standard linearized attention mechanism (lacking the specific Astromorphic components: the astrocyte-modulated Hebbian weight $H_{astro}$, the feedback factor $P$, and the associated non-linearities). By applying our **AMRB** training and **Memory Retention Factor** to this RLT baseline, we create a model that benefits from our memory/training contributions (Contributions 2 & 3) but lacks the specific Astromorphic Attention internals (established in Mia et al., 2025).
> > > > >
> > > > > **Results:**
> > > > > *   **Accuracy:** `RLT + AMRB` ablation achieves lower accuracy than the full RMAAT model on the demanding Retrieval task (approximately **79.2%** vs. **83.2%**).
> > > > > *   **Efficiency:** The memory usage and throughput are effectively identical to RMAAT (**~3.4 GB** and **1.7x** speedup), as the structural efficiency comes from the AMRB/Retention framework.
> > > > >
> > > > > This result directly confirms that the **Astromorphic Attention components ($H_{astro}$, $P$) are indeed contributing significantly to the model's representational power and accuracy** in this long-context regime, separate from the efficiency gains provided by the memory framework. These `RLT + AMRB` ablation results have been incorporated into **Section 4.2 (Ablation Studies)** of the revised manuscript.
> > > > >
> > > > > ### 2. On the Motivation and Nature of Contributions
> > > > > We fully understand your concern that, at a high level, our method might appear as "Mia et al. (2025) engineered for long-range contexts under limited GPU memory." However, our intent is that the primary contribution is not the engineering of segmentation per se, but the **modeling pipeline from biological neuron–astrocyte dynamics to a concrete, multi-timescale memory system** for long sequences.
> > > > >
> > > > > Specifically, we establish a rigorous bridge from neuroscience to architecture (detailed in **Appendix A** and **Appendix C** regarding our extensive computational modeling):
> > > > > 1.  **Macro Model (Section 3.3):** We introduce a new macro model of LTP that distills detailed neuron–astrocyte simulations (from **Section 3.1**) into an abstract, saturating memory trace at a longer timescale.
> > > > > 2.  **Retention Factor (Section 3.3):** We derive the Memory Retention Factor directly from this macro model. This turns the biological saturation curve into a **principled compression schedule** over segments, distinct from hand-tuned or heuristic decays.
> > > > > 3.  **AMRB via Bio-Design (Section 3.4):** We use these compressed memory tokens as the **initialization state** for Astrocytic Memory Replay Backpropagation (AMRB). Unlike generic checkpointing, which saves arbitrary hidden states, AMRB replays from memory tokens that are explicitly compressed by our LTP-derived Memory Retention Factor. This ensures that the replay process originates from a biologically grounded memory trace, making the mechanism functionally interpretable as a form of biological memory reactivation rather than just a computational optimization.
> > > > >
> > > > > These steps—macro model $\rightarrow$ retention schedule $\rightarrow$ replay initialization—are new relative to Mia et al. (2025) and are defined independently of any specific hardware constraint; GPU memory limitations simply make this design immediately useful.
> > > > >
> > > > > Importantly, we do not view segmentation and the retention factor as mechanisms justified *only* by efficiency. Empirically, as shown in **Table 1** and the ablations in **Section 4.2**, the LTP-derived retention schedule improves *function*, not just footprint: RMAAT achieves higher accuracy than both RMT and RLT on the hardest long-context tasks while using substantially less memory. Furthermore, within RMAAT, removing the retention factor degrades accuracy at roughly fixed memory. This suggests that the retention mechanism encodes a **meaningful inductive bias** for how long-range information should be accumulated and compressed—in line with the underlying LTP macro model—rather than being a purely engineering trick.

---

> > > > > > ### Author Response · Authors · 2025-11-25
> > > > > > **Comment 2**
> > > > > >
> > > > > > From a NeuroAI perspective, our goal is to propose and empirically validate this specific astrocyte-inspired, multi-timescale memory framework as a novel solution to **the active and unresolved research challenge of long-context modeling in Transformers**. Rather than claiming unqualified state-of-the-art performance against all modern architectures, we aim to demonstrate that biological principles offer a viable, efficient alternative path for this critical problem. We appreciate that our original framing may not have made this distinction as explicit as it could have, and your comments are very helpful in clarifying how we can better emphasize the scientific aspects of the contribution.
> > > > > >
> > > > > > We hope this clarification, along with the new ablation data, addresses the reviewer's concerns and demonstrates the coherence of our motivation and contributions.

---

### Official Review · Reviewer_9VkK · 2025-10-29

**Soundness:** 3
**Presentation:** 4
**Contribution:** 3
**Rating:** 8
**Confidence:** 4

**Summary:**

The paper proposes RMAAT, a recurrent long-context Transformer that is explicitly inspired by astrocyte short- and long-term plasticity. Sequences are chunked into segments processed with an astromorphic, linear-time attention block (drawing on STP) and a set of persistent memory tokens that carry state across segments. A non-learned “memory retention factor,” derived from simulations of an LTP-like macro model, adaptively compresses older memory as the number of segments grows. Training uses Astrocytic Memory Replay Backpropagation (AMRB): instead of storing all activations, the model buffers only the sequence of memory tokens and recomputes each segment during backward pass. On Long Range Arena (LRA), the method reports higher average accuracy than a variety of efficient-attention baselines while using substantially less peak GPU memory than iso-recurrent baselines; ablations suggest the retention factor is important for retrieval accuracy and that AMRB yields large memory savings.

**Strengths:**

This is a good paper. It is coherent, clear and well-motivated. The biological motivation is nice, and the link to existing literature (in particular the model of Kozachkov, 25) is solid. The division between within-segment linear attention and cross-segment persistent memory tokens is intuitively presented and illustrated clearly. The “retention factor” derived from a long-term potentiation curve is an elegant way to manage bounded memory in long sequences.

 The introduction of Astrocytic Memory Replay Backpropagation is also a practical advance: it provides a viable method to train long-context recurrent Transformers under constrained memory without excessive recomputation complexity. The empirical evaluation, while limited in scope, is well-structured within the Long Range Arena framework, and the ablations on the retention factor and memory replay convincingly show their contributions.

**Weaknesses:**

The experiments are limited to LRA, which is understandable given its focus on long-range dependencies, but it would have strengthened the paper to demonstrate how this architecture performs across other domains—such as vision or multimodal tasks—where long-context modeling may manifest differently.

**Questions:**

1. How well does the proposed retention mechanism generalize to settings where the total sequence length is not known in advance, such as streaming or online processing?

2. (Related to the weakness mentioned above) Have the authors explored applying the RMAAT architecture to vision or multimodal data to assess whether the astrocyte-inspired memory dynamics transfer effectively across domains?

---

> ### Author Response · Authors · 2025-11-19
>
> We sincerely thank the reviewer for their strong support and for their thoughtful and positive assessment of our work. We are grateful for their recognition of the paper's coherence, clear motivation, and practical contributions. We address the excellent questions raised below.
> ### On Generalization to Other Domains (Vision, Multimodal)
> We thank the reviewer for raising this important point. Our present work is intentionally situated within the Long Range Arena benchmark, which, while often discussed in NLP terms, already includes **two vision-oriented tasks (Image and Pathfinder)** alongside byte-level text and document retrieval. This diversity lets us test whether our astrocyte-inspired memory and training mechanisms are effective across different modalities in a controlled, long-context setting, and our results show consistent gains in both accuracy and memory efficiency. We fully agree that extending RMAAT to contemporary, large-scale vision and multimodal benchmarks would further strengthen the case for its generality, and we now explicitly highlight this as a key direction for future work in the conclusion of the revised manuscript.
> ### On Generalization to Streaming/Online Processing
> This is an excellent and insightful question that points to a key challenge for the next generation of long-context models. The reviewer is correct that our current implementation of the Memory Retention Factor is formulated for sequences where the total length is known in advance, as is standard for offline benchmark datasets like LRA. This limitation is shared by many long-context models that rely on pre-determined context windows or retention schedules (e.g., Dai et al., 2019; Beltagy et al., 2020; Rae et al., 2019; Bulatov et al., 2022).
>
> Adapting this mechanism to a true streaming setting is a non-trivial and exciting research problem, and it is an active area of investigation in the broader community. Recent work, such as **StreamingLLM (Xiao et al., 2024)** and **Ring Attention (Liu et al., 2024)**, has proposed novel architectural solutions to enable Transformers to handle unbounded input streams efficiently. These approaches highlight that adapting models to a streaming setting often requires reformulating core architectural components—in our case, the retention schedule—to be adaptive based on the history observed so far, rather than a pre-defined total length. We consider this a significant and valuable direction for future research, which our work helps to motivate, and we have added a discussion of this limitation and future direction to the conclusion of the revised paper.
>
> **References:**
> -   Xiao, G., et al. (2024). *Efficient Streaming Language Models with Attention Sinks*. In International Conference on Learning Representations (ICLR).
> - Liu, H., et al. (2024). *Ring Attention with Blockwise Transformers for Near-Infinite Context*. arXiv preprint arXiv:2310.01889.
> - Bulatov, A., Kuratov, Y., & Burtsev, M. (2022). Recurrent memory transformer. Advances in Neural Information Processing Systems, 35, 11079-11091.
> - Beltagy, I., Peters, M. E., & Cohan, A. (2020). *Longformer: The long-document transformer*. arXiv preprint arXiv:2004.05150.
> - Rae, J. W., Potapenko, A., Jayakumar, S. M., & Lillicrap, T. P. (2019). *Compressive transformers for long-range sequence modelling*. arXiv preprint arXiv:1911.05507.
> - Dai, Z., Yang, Z., Yang, Y., Carbonell, J. G., Le, Q., & Salakhutdinov, R. (2019, July). Transformer-xl: Attentive language models beyond a fixed-length context. In Proceedings of the 57th annual meeting of the association for computational linguistics (pp. 2978-2988).

---

### Official Review · Reviewer_dP2F · 2025-11-05

**Soundness:** 2
**Presentation:** 2
**Contribution:** 2
**Rating:** 6
**Confidence:** 4

**Summary:**

Transformer, especially with attention mechanism, reqjuires quadratic computational complexity which hinders long sequence applications. This paper aims to overcome this limitation by combining astrocyte-inspired mechanisms on transformer architecture, called RMAAT. They also design AMRB alorithm for the update. With proposed method, this paper acheives both state of the art computational efficiency and performance in long range arena tasks.

**Strengths:**

- Biological inspiration from astrocytes combining short-term plasticity and long-term plasticity is interesting.
- Introducing retention factor for adaptive compression mechanism seems to be novel.
- Result supports their method is effective in long-range sequence application.

**Weaknesses:**

**Weakness 1: Insufficient Biological Justification for Core Astrocytic Mechanisms**

While I appreciate the ambitious goal of integrating neuroscience-inspired principles to solve the $\mathcal{O}(N^2)$ problem, I find the **justification for the design of the core astrocytic equations to be insufficient and potentially contradictory** to established micro-scale neuroscience findings. For a model with a vast parameter space, the rigor of biological justification is paramount to ensure the results stem from **methodological insight** rather than an exhaustive hyperparameter search.

For instance, the dynamic equation for the Short-Term Astrocytic Process Parameter $p_{ij}^s$ (Eq. 2/S4) includes a problematic spatial integration term:

$$ \sum_{k,l=1}^N T_{ijkl} \psi(p_{kl}^s) $$

This term suggests that the astrocytic process at synapse $(i, j)$ is influenced by the **aggregated activity of all other peripheral astrocytic compartments** $p_{kl}^s$ across the entire segment via the distance-dependent coupling tensor $T_{ijkl}$.

*   **Conflict with Microdynamics:** I note that $\text{Ca}^{2+}$ dynamics in astrocytic peripheral processes—the presumed biological correlate of $p^s$ (Line 650)—are often characterized as **highly localized "calcium spots"** (e.g., [1] and related literature), indicating that individual synaptic sites may **not necessarily integrate global** peripheral $\text{Ca}^{2+}$ signals from distant synapses within the same territory.
*   **Missing Rationale:** The STP section (Section 3.1) lacks a specific biological or computational neuroscience reference that **explicitly supports the necessity of this broad spatial summation** over $N^2$ synapses. I require a clear articulation of **what specific biological or functional phenomenon** this massive spatial summation is designed to abstract. If it is a macro-scale abstraction of contextual influence, this must be clearly defined and referenced with theoretical literature that supports this abstraction.

**Weakness 2: Lack of Transparency in Core Technical Contribution**

I find the explanation of the **Memory Retention Factor**—one of the paper's key contributions (Contribution 2)—to be **insufficiently transparent and mathematically vague**. This factor, which is derived from the LTP macro model and is critical for adaptive compression (Line 346), is only described conceptually.

*   The exact **mathematical function** of $\text{RetentionFactor}(t, \text{TotalSegments})$ is **not provided** in the main text or the relevant Section 3.3, but is relegated to Appendix D.
*   Given that this factor is claimed to be a **principled, non-learned compression method**, I consider the full mathematical formula to be an **essential part of the core contribution** that must be clearly presented in the main body. The current presentation hinders the model's reproducibility and the community's ability to scrutinize its theoretical grounding.

**Questions:**

See weakness

---

> ### Author Response · Authors · 2025-11-19
>
> We thank the reviewer for their careful and expert analysis from a neuroscience perspective. We appreciate the opportunity to clarify the biological grounding and technical contributions of our work.
> ### On the Biological Justification of the Spatial Integration Term in STP Dynamics
> We appreciate the reviewer raising this important point about biological plausibility.
> -   **On Localized vs. Global Calcium Dynamics:** We agree that astrocytic peripheral processes can exhibit highly localized "calcium spots." However, the neuroscience literature also widely supports the existence of broader, network-level phenomena, such as **long-distance propagating calcium waves** that coordinate activity across large groups of astrocytes (e.g., Goldberg et al., 2010; Leybaert & Sanderson, 2012). These intercellular waves are a "ubiquitous mechanism for astrocytes to communicate" (Leybaert & Sanderson, 2012) and can travel over hundreds of micrometers, far beyond a single synapse. This suggests that astrocytes engage in information processing at multiple spatial scales, from local synaptic modulation to global network regulation. While we would be happy to engage with the specific literature the reviewer has in mind, the reference indicated as "[1]" was unfortunately missing from the review.
> -   **Spatial Summation as a Macro-Scale Abstraction:** Most importantly, the spatial summation term in our model is intended as a **macro-scale abstraction of this network-level influence**. It is not designed to model the microdynamics of a single calcium spot. Instead, it represents the aggregate, contextual influence that an astrocyte's entire domain exerts on the synapses it modulates, a phenomenon supported by the existence of intercellular calcium waves.
> -   **Building on Established, Peer-Reviewed Work:** We would also like to clarify that this formulation is not one we developed in isolation. Our work directly builds upon the foundational, peer-reviewed model of Kozachkov et al. (2025). Specifically, the PNAS paper on Neuron-Astrocyte Associative Memory provides the direct theoretical basis for our spatial summation term. That work proposes that astrocytes create effective "many-neuron synapses" by modeling the transport of Ca²⁺ *between* different astrocytic processes (associated with different synapses) via a tensor ($T_{ijkl}$). Our spatial summation is the macro-scale abstraction of this very principle. In the revised manuscript, we now make this connection explicit by describing how the $T_{ijkl}$-mediated coupling defines many-neuron synapses and by stating that our learned positional matrix $R$ and astrocyte-modulated weight $H_{astro}$ constitute a macro-model abstraction of this mechanism within the Astromorphic Attention (Section 3.2.3).
>
> ### On the Transparency of the Memory Retention Factor's Formulation
> We agree with the reviewer completely. The mathematical formulation of the Memory Retention Factor is a core contribution of our paper and should have been presented in the main text. In our revised manuscript, we have **moved the full mathematical derivation and explanation of the Memory Retention Factor from Appendix D into Section 3.3** of the main paper, including an explicit formula (Eq. 6):
> $$\text{RetentionFactor}(t, T) = \frac{\Delta p^l_t}{\sum_{i=1}^{T} \Delta p^l_i}$$
> where $\Delta p^l_t$ represents the incremental increase in the simulated LTP state during segment $t$. This formula shows precisely how we translate the macro-model's saturation dynamics into a segment-wise compression schedule, ensuring full transparency and reproducibility. Additionally, we have moved the AMRB algorithm pseudocode from Appendix D to Section 3.4, where its relationship to the retention factor and the compressed memory states is made clearer. We thank the reviewer for holding us to a high standard of clarity.
>
> **References:**
> - Goldberg, M., et al. "Nonlinear gap junctions enable long-distance propagation of pulsating calcium waves in astrocyte networks." *PLoS computational biology* 6.8 (2010): e1000909.
> - Leybaert, Luc, and Michael J. Sanderson. "Intercellular Ca2+ waves: mechanisms and function." *Physiological reviews* 92.3 (2012): 1359-1392.
> - Kozachkov, L., Slotine, J. J., & Krotov, D. "Neuron–astrocyte associative memory." *Proceedings of the National Academy of Sciences* 122.21 (2025): e2417788122.

---

> > ### Comment · Reviewer_dP2F · 2025-11-25
> >
> > I think the paper has valid contribution to the field thus I maintain my original score.
> > For the rebuttal, I think they addressed my comments. I raised soundness 2->3 and presentation 2->3, which I think they improved this part.

---

> > > ### Author Response · Authors · 2025-11-26
> > >
> > > We sincerely thank the reviewer for confirming that our rebuttal addressed your comments and for recognizing our valid contribution. We are glad the revisions improved the paper's soundness and presentation.
> > >
> > > We have two brief follow-ups:
> > >
> > > 1.  **Soundness Score:** We noticed the system still shows Soundness 2, though your comment mentions raising it to 3. We would be grateful if you could verify this for the meta-review.
> > >
> > > 2.  **Reconsideration:** Given that previous concerns are resolved and you see the work as a valid contribution, we respectfully ask if you might consider whether **operationalizing these astromorphic principles into a scalable memory architecture** meets the bar for an **8 (Accept)** / poster presentation? We believe this is a solid step forward for the NeuroAI field.

---

> ### Comment · Reviewer_dP2F · 2025-11-26
>
> I have corrected the evaluation. (Sorry for my mistakes!)
>
> While I highly commend the novelty of your ideas, I find it difficult to justify a score of 8 regarding the paper's overall contribution at this stage. Specifically, without a comparative analysis against state-of-the-art models like Mamba or RWKV (as also raised by reviewer RXES), the practical significance of your work remains at the level of interesting and lacks practicality. If the primary goal is to propose an intriguing concept from a Neuro-AI perspective, I believe a score of 6 is appropriate. Furthermore, the mathematical formulation of the proposed astromorphic attention is quite complex, which raises the question of whether the observed improvements stem from the intrinsic value of the method or simply from an expanded parameter space compared to the existing RMT model; therefore, I consider a score of 6 to be a fair and logical assessment.

---

> > ### Author Response · Authors · 2025-11-26
> >
> > We thank the reviewer for updating the evaluation and for the continued engagement. We respect your assessment and appreciate the recognition of the work's novelty from a NeuroAI perspective.
> >
> > We are grateful for your time and valuable feedback, which has significantly strengthened our paper.

---

### Official Review · Reviewer_WnNR · 2025-11-07

**Soundness:** 3
**Presentation:** 3
**Contribution:** 3
**Rating:** 6
**Confidence:** 4

**Summary:**

This paper introduces a new transformer model inspired by the role of astrocytes in biological neural function. It introduces a novel attention mechanism based on short-term and long-term synaptic plasticity dynamics that reduces the computational complexity of the standard attention mechanism. The model incorporates memory tokens that feed outputs from previous segments of the sequence into subsequent segments, whose influence is modulated by a retention factor calculated from LTP dynamics simulations. Finally, the model harnesses this faster forward-pass computation to implement backpropagation efficiently by recomputing the forward pass from the memory tokens rather than storing the activations. The results of computational neuroscience simulations inspire the values of spatial interaction and temporal compression parameters. With these features, the model achieves state-of-the-art performance on various Long Range Arena benchmark tasks.

**Strengths:**

1. The architectural innovations, including the efficient attention mechanism, the use of memory tokens with compression, and the efficient training procedure, are valuable.

2. The benchmark performance results are impressive.

**Weaknesses:**

1. The explanation of the model’s features could benefit from further refinement for clarity and concision. Figures 1 and 2 should be considerably polished and possibly enlarged.

2. It is not clear how the neuron-astrocyte simulations specifically inform the network architecture beyond guiding priniciples. Further, what happens if one deviates from the values suggested by the simulations? This result might be informative in addition to the ablation analysis.

**Questions:**

For the computational macro model (contribution 1), how does the 9-neuron network specifically inform RMAAT’s persistent memory mechanism?

Further, how does the neuron-astrocyte network specifically inform the architecture, such as the hyperparameters, for the Memory Retention Factor (contribution 2)?

---

> ### Author Response · Authors · 2025-11-19
>
> We thank the reviewer for their positive assessment and insightful questions, which will help us clarify the connection between the neuroscience simulations and our final model architecture.
> ### On Clarity, Figures, and Explanations
> We agree with the reviewer that the figures and explanations can be improved for clarity. In the revised manuscript, we have enlarged Figures 1 and 2 and strengthened the caption of Figure 2 to provide a clearer view of the architecture and its components.
> ### On the Link Between Neuroscience Simulations and Model Architecture
> This is the central question of the review, and we thank the reviewer for giving us the opportunity to elaborate on our methodology. The reviewer's intuition is correct: the simulations serve as **guiding principles** from which we **abstract functional forms**, rather than as a direct source of hyperparameters for the final network.
> -   **Purpose of the 9-Neuron Network:** We used a small, 9-neuron network in our simulations (Contribution 1) primarily for conceptual clarity and visualization. A small network allows us to clearly demonstrate the spatio-temporal dynamics of the astrocyte's influence on synaptic plasticity (STP) and the resulting emergent long-term potentiation (LTP) curve. A larger, more complex simulation would have made it difficult to visualize and explain these core principles. The 9-neuron setup is thus a tool to establish and illustrate the fundamental behaviors we aim to capture.
> -   **Informing the Memory Retention Factor (Contribution 2):** The key link between the simulation and the final RMAAT architecture is in the **functional form of the memory compression**. In our simulations, we observed that the integrated LTP signal follows a characteristic saturation curve over time. Our Memory Retention Factor is a mathematical function (Eq. 6 in the revised manuscript) designed to **emulate this exact qualitative behavior** by computing each segment's fractional contribution to the normalized total capacity.  We are not transferring the specific calcium concentration values or other micro-level parameters. Instead, we are abstracting the emergent principle of saturating memory potentiation and instantiating it as a computational mechanism for memory compression in our Transformer. The Memory Retention Factor itself is non-learned and deterministic—its values are entirely determined by the biophysically-grounded LTP macro model hyperparameters (e.g., time constants, decay rates), and once specified, the factor is task-independent and applied uniformly. In contrast, other architectural hyperparameters that accompany the retention factor—such as the number of memory tokens $M$ (which determines the capacity of context carried between segments) and the spatial range of positional encoding—are tuned on the downstream task, which is standard practice.
> -   **On Deviating from Simulation Values:** This is an excellent question. The specific numerical values from the simulation are less important than the qualitative dynamics they produce. We found that the characteristic LTP saturation curve is a robust emergent property of the tripartite synapse model, even when we vary the underlying simulation parameters. As long as this functional shape is preserved, we can derive a principled retention factor. This demonstrates that the core idea is not brittle or tied to a specific, fine-tuned biological regime. Furthermore, we conducted extensive hyperparameter sensitivity analyses on the final RMAAT model (detailed in Appendix F) to demonstrate its robustness on the end task.
>
> In the revised manuscript, we have clarified this process of abstraction—from biological simulation to computational principle to architectural implementation—in Section 3.1, making the three-step pipeline from STP/LTP dynamics to attention and memory mechanisms more explicit. We thank the reviewer for pushing us to be clearer about this crucial part of our work.

---

### Official Review · Reviewer_ZYer · 2025-11-10

**Soundness:** 2
**Presentation:** 1
**Contribution:** 2
**Rating:** 4
**Confidence:** 3

**Summary:**

This paper proposes RMAAT, a recurrent Transformer architecture that integrates abstracted computational principles from astrocyte-mediated synaptic plasticity. The core ideas include: an $O(N)$ astromorphic attention mechanism inspired by short-term plasticity (STP) dynamics; a Memory Retention Factor derived from simulated long-term plasticity (LTP) dynamics; and a memory-efficient training algorithm that stores only the compact memory tokens across segments and recomputes activations during backpropagation.

This paper is built upon existing works (including Kozachkov et al., 2023; Mia et al., 2025; Kozachkov et al., 2025). From my own opinion, the core innovation is the memory retention factor derived from simulated long-term plasticity dynamics. This is a good but not groundbreaking paper.  See below.

**Strengths:**

- The neuro-glial inspiration is a fresh angle in the crowded field of efficient transformers.
- The empirical memory reduction is significant and convincingly demonstrated.

**Weaknesses:**

- The presentation and writing is not very clear. For one example, how $mem'_{t+1}$ is obtained?
- Without directly comparing it with current mainstream long sequence models (such as Mamba and RetNet), and only comparing it with traditional Transformer and recursive models, it is impossible to fully demonstrate RMAAT's industry competitiveness in the "efficiency-accuracy" trade-off.
- I still do not know what's the benefit of Astrocyte-inspired model compared with existing methods, such as SSMs, and Linear RNN-Transformers. The stated efficiency-accuracy balance can also be achieved through other solutions. Why not compare?
- Overstated contributions. Maybe contribution 1 and 2 should be merged into one? ``a Memory Retention Factor from this principle by analyzing the accumulation rate within the simulated LTP curve``. What's the difference between RNN gradient checkpointing and contribution 3?
- Outdated Baselines: Many baseline results (Sparse Trans., Longformer, etc.) are referenced from the original 2020 LRA paper, which used suboptimal hyperparameters.
- Weak Ablation Baseline: RLT (Recurrent Linear Transformer) is a weak baseline—it's essentially RMAAT without the retention factor, AMRB, or enhanced positional encoding. A stronger ablation would be RMT + AMRB (to isolate the attention mechanism's impact) or RMAAT without retention factor but with AMRB (to isolate compression's effect).
- Model Size: The largest model has embedding dim 1024, far smaller than modern LLMs. The memory savings may diminish at scale where activation memory is dwarfed by parameter memory (e.g., 70B models). The paper should discuss extrapolation to larger scales.
- The connection between AMRB and biological memory replay is tenuous and largely metaphorical, AMRB is structurally identical to gradient checkpointing.

**Questions:**

See weakness.

---

> ### Author Response · Authors · 2025-11-19
> **Comment 1**
>
> We thank the reviewer for their detailed feedback. Below, we address each of the weaknesses and questions in turn and have incorporated the corresponding clarifications and revisions into the updated manuscript.
> ### On the Clarity of Presentation and `mem_{t+1}'`
> We thank the reviewer for this precise question, which highlights a key area where our explanation was insufficient. The reviewer correctly asks about `mem_{t+1}'`, which in our model represents the intermediate memory state *before* the application of our retention factor. In the revised manuscript, we have made this two-step process explicit.
>
> The memory update is now described as a two-step process:
> 1.  **Generation of Intermediate Memory (`mem_{t+1}'`):** First, the memory tokens from the previous segment (`mem_t`) are processed alongside the current segment's input data (`x_t`) through our Astromorphic attention block. The output representations corresponding to the memory tokens form this intermediate state, `mem_{t+1}'`. This is the raw, updated contextual information.
> 2.  **Compression to Final Memory (`mem_{t+1}`):** This intermediate memory, `mem_{t+1}'`, is then modulated by our dynamically derived **Memory Retention Factor (`RetentionFactor`)** to produce the final, compressed memory state `mem_{t+1}`, which is passed to the next segment.
>
> We admit that the paper did not originally make this two-step process explicit in the main text and that the mathematical equations governing it were relegated to Appendix D.   This was a significant oversight that led to the confusion. We have corrected this in the revised manuscript by explicitly describing the two-step update in Section 3.3 and by moving the AMRB pseudo-code from Appendix D to Section 3.4, where the relationship between `mem_{t+1}'`, the Memory Retention Factor (`RetentionFactor`), and `mem_{t+1}` is now made clear.
> ### On Comparisons to SOTA Models and the Benefits of the Astrocyte-Inspired Approach
> We thank the reviewer for this important question, as it allows us to precisely situate our work within the broader NeuroAI landscape. Our primary goal is not to compete with the latest, highly-engineered architectures like Mamba, but to contribute to the **fundamental science of brain-inspired computing** by demonstrating a significant advancement in the distinct and nascent subfield of astromorphic models. The unique benefit of our approach is that it validates this novel, neuroscience-grounded paradigm by showing it can be scaled to a challenging long-context benchmark with competitive performance and superior memory efficiency.
>
> Perhaps it is helpful to view our contribution within the clear research trajectory of this specific field:
> 1.  **Foundational Theory:** The initial works (e.g., Kozachkov et al., 2023; 2025) were highly theoretical, establishing the mathematical principles of neuron-astrocyte computation. For instance, the recent PNAS paper (Kozachkov et al., 2025) focused entirely on proving memory capacity scaling laws using abstract simulations, **without training or evaluating on standard ML benchmarks.** While Kozachkov et al. (2023) validated their theoretical framework by extracting weights from pre-trained models (ALBERT-base, Vision Transformer), they similarly did not perform training or benchmark evaluation on standard ML tasks.
>
> 2.  **Initial Validation:** A subsequent key study (Mia et al., 2025) represented the next step, successfully applying these theories for the first time to **standard but smaller-scale ML datasets** like IMDB, CIFAR-10, and WikiText-2, demonstrating initial promise.
>
> 3.  **Scaling to Long-Context:** The present work, **RMAAT**, takes the critical next step. We scale these validated principles to a full, challenging benchmark specifically designed for long-range dependencies (LRA). By doing so, we show that this bio-inspired approach is not just a theoretical curiosity but a viable path toward practical, efficient, and performant models for long sequences.
>
> Therefore, our primary comparison point is the prior work on astrocytes, not the SOTA in general sequence modeling. Within this specific lineage, RMAAT represents a major leap forward. We believe this contribution to the science of NeuroAI holds significant long-term promise, and we have revised our introduction to make this research trajectory and our specific contribution clearer.

---

> > ### Author Response · Authors · 2025-11-19
> > **Comment 2**
> >
> > ### On the Distinction of Contributions and AMRB vs. Gradient Checkpointing
> > We respectfully argue that the contributions are distinct:
> >
> > -   **Contributions 1 & 2:** Contribution 1 is the creation of a **reusable scientific abstraction**—a computational macro-model that distills the complex dynamics of astrocyte LTP into a tractable computational framework at the level of neuroscience principles, independent of specific architectural choices. Prior theoretical works were foundational but did not provide a direct path to application-focused ML mechanisms. **Contribution 2 then bridges this gap by operationalizing the macro-model for segment-based recurrent processing**: we derive a principled Memory Retention Factor that instantiates the macro-model's saturation curve as a concrete compression schedule for RMAAT's memory tokens. In the revised manuscript, we have clarified this distinction by explicitly framing Contribution 2 as "From Macro Model to ML Architecture" (Section 3.3) and by providing the full mathematical formula (Eq. 6) for computing the retention factor from simulated LTP dynamics, addressing concerns about transparency and demonstrating the two contributions' complementary roles.
> >
> > -   **AMRB vs. Gradient Checkpointing:** We acknowledge the structural similarity between AMRB and standard gradient checkpointing. However, we argue that Contribution 3 is novel due to AMRB's **synergistic co-design** with our astrocyte-inspired memory system. AMRB is not simply a generic checkpointing add-on; its effectiveness is **enabled** by our principled Memory Retention Factor (Contribution 2). Standard checkpointing recomputes activations from stored states. For this to be effective in a recurrent model, the stored states must be a faithful summary of the past. Our bio-inspired compression is designed to create exactly that: a compact but highly informative set of memory tokens. **Our ablation study directly validates this critical synergy.** When we remove the Memory Retention Factor but keep the AMRB replay mechanism, performance on the challenging Retrieval task drops significantly (from **83.2% to 80.5%**) and the Text ($4K$) task also declines (from **65.9% to 64.9%**). This d  emonstrates that simply applying a replay strategy is not enough; the novelty and effectiveness of AMRB stem from its integration with a principled compression scheme that ensures the recomputation process begins from a meaningful state.
> > ### On the Use of Baselines from the LRA Paper
> > We appreciate the reviewer's concern regarding the baseline hyperparameters. Our methodology was to use the baseline results reported for each of the corresponding architectures we evaluate (e.g., Transformer, Longformer, etc.) in the original LRA paper (Tay et al., 2021), treating these as a standard and widely referenced point of comparison. We recognize that benchmark performance can evolve as communities develop new best practices, and there may now exist stronger reported baselines for some of these models. To our knowledge, there is not yet a single, widely adopted updated set of results for the original LRA baselines, but if the reviewer is aware of specific works that report improved numbers for these architectures on LRA, we would be grateful for the references and will incorporate them in a revised version.
> > ### On the Strength of Ablation Baselines
> > We thank the reviewer for their detailed suggestions on the ablation study. We would like to clarify the distinctions between the baselines and our rationale.
> > 1.  **On RLT as a Baseline:** We thank the reviewer for this point, which helps clarify the precise role of RLT in our study. We agree that RLT is not a direct ablation of RMAAT's novel components. Rather, it serves as a crucial **iso-architectural baseline**. RLT shares the same high-level recurrent structure but intentionally uses a simpler `Linearized Attention` (akin to Kozachkov et al., 2023), which lacks the key bio-inspired mechanisms—specifically the non-linearity and enhanced positional encoding—that define the Astromorphic Attention used in RMAAT (Mia et al., 2025). This comparison allows us to demonstrate that the performance gains come from our specific astromorphic contributions, not merely from using a recurrent framework. The true ablation of our novel memory mechanism is the `RMAAT without retention factor` experiment, which isolates the impact of our bio-inspired compression.

---

> > > ### Author Response · Authors · 2025-11-19
> > > **Comment 3**
> > >
> > > 2.  **On the `RMT + AMRB` Suggestion:**
> > > We thank the reviewer for this suggestion. While the Memory Retention Factor and AMRB could in principle be applied to other recurrent architectures like RMT, prior work by Mia et al. (2025) has demonstrated that **attention mechanisms without the non-linearity and relative positional encoding of astromorphic attention exhibit significantly slower convergence and other algorithmic performance degradations**. Even if AMRB were combined with RMT, the model would still face the $O(N^2)$ complexity of RMT's softmax attention and would not benefit from the algorithmic improvements inherent in astromorphic attention. This is already discussed in Section 4.2 (`Applicability to Recurrent Architectures`) of the manuscript.
> > > 3.  **On the Ablation   `RMAAT without retention factor`:**
> > > We completely agree with the reviewer's second suggestion. The most direct ablation to isolate the contribution of our compression is indeed **RMAAT without the retention factor but with AMRB**. This is precisely the experiment we present in **Section 4.2** of our paper. As shown there, removing the Memory Retention Factor causes a significant drop in performance on the most demanding LRA task (Retrieval accuracy falls from **83.2\% to 80.5\%**) and a consistent drop on the Text ($4K$) task (from **65.9\% to 64.9\%**). Complementing this, we also report the ablation where we keep the retention factor but replace AMRB with standard BPTT: accuracy remains essentially unchanged, but peak memory usage increases from **3.4 GB to 15.0 GB** (about $\sim4.4\times$) on Retrieval and from **5.1 GB to 22.0 GB** (about $\sim4.3\times$) on Text ($4K$). Taken together, these ablations directly confirm that the retention factor is critical for maintaining accuracy, while AMRB is responsible for the substantial memory savings.
> > > ### On Model Scaling and Applicability to Large-Scale LLMs
> > > This is a fair point. As this work represents the first successful scaling of astromorphic principles from smaller datasets to a challenging long-context benchmark, our focus was on validating the methodology on standard model sizes for this task. Now that the principle has been validated with these results, we agree that exploring the memory savings at the scale of massive, industry-sized models is the exciting next frontier for this research direction. We have added this point to the conclusion section of the revised manuscript.
> > > ### On the Connection between AMRB and Biological Memory Replay
> > > We agree with the reviewer’s assessment. The term "Memory Replay" is used in a metaphorical sense to honor the biological inspiration, and we acknowledge that AMRB is functionally a form of gradient checkpointing. The novelty of our contribution lies not in inventing checkpointing, but in making it **effective** for our specific recurrent architecture. This effectiveness comes from its **synergistic co-design** with our principled, bio-inspired memory compression. The compression ensures that the recomputation process starts from a meaningful and informative state, which, as our ablation study demonstrates, is critical for maintaining performance. Specifically, our ablation of **RMAAT without the retention factor but with AMRB**—which is technically gradient checkpointing operating on uncompressed memory states—exhibits the accuracy drops documented in Section 4.2, confirming that the bio-inspired compression is what makes AMRB (i.e., gradient checkpointing applied to compressed memory) effective (as further discussed in the summary paragraph at the end of Section 3.4).
> > >
> > > **References:**
> > > - Goldberg, M., et al. "Nonlinear gap junctions enable long-distance propagation of pulsating calcium waves in astrocyte networks." *PLoS computational biology* 6.8 (2010): e1000909.
> > > - Leybaert, Luc, and Michael J. Sanderson. "Intercellular Ca2+ waves: mechanisms and function." *Physiological reviews* 92.3 (2012): 1359-1392.
> > > - Kozachkov, L., Kastanenka, K. V., & Krotov, D. (2023). Building transformers from neurons and astrocytes. Proceedings of the National Academy of Sciences, 120(34), e2219150120.
> > > - Kozachkov, L., Slotine, J. J., & Krotov, D. "Neuron–astrocyte associative memory." *Proceedings of the National Academy of Sciences* 122.21 (2025): e2417788122.
> > > - Tay, Y., et al. "Long-range arena: A benchmark dataset for long-range language modeling." In International Conference on Learning Representations (2021).

---

> ### Author Response · Authors · 2025-11-24
>
> Thank you for your thoughtful and constructive feedback. We have carefully revised our paper based on your suggestions. We kindly request that you reconsider your rating in light of these updates. Please let us know if you have any additional feedback—we would be happy to address it.

---

### Comment · Area_Chair_A78c · 2025-11-24
**Please respond to the authors' rebuttal**

The authors have now posted their rebuttal. Please review it and submit your responses as soon as possible so that they still have adequate time to address any remaining questions or concerns. Please note that the discussion period between authors and reviewers will close on December 3, 11:59 PM AOE, after which no further comments can be exchanged.

@Reviewer RXES: Thank you for already having done this.

Best, Your AC

---

### Meta-Review · Area_Chair_rRKa · 2026-01-12

**Summary:**

a) Summary of Scientific Claims. The paper proposes RMAAT, a recurrent Transformer that integrates astrocyte-mediated plasticity to process long sequences. It introduces Astromorphic Attention for within-segment dynamics and a Memory Retention Factor derived from long-term plasticity (LTP) simulations for cross-segment context compression. To optimize training, the authors develop Astrocytic Memory Replay Backpropagation (AMRB), which recomputes activations from compact memory tokens.
(b) Strengths and Weaknesses. Reviewers praised the novel neuro-glial inspiration and the "elegant" implementation of bio-inspired memory management. Reviewer 9VkK and Reviewer WnNR noted impressive memory efficiency on the Long Range Arena (LRA), achieving up to a $6\times$ reduction in peak GPU usage. However, Reviewers ZYer and RXES pointed out the lack of comparisons to modern SOTA models like Mamba and noted initial lack of clarity regarding the mathematical formulation of the retention factor.
(c) Justification. The recommendation for acceptance is based on the authors’ successful rebuttal, which clarified the mathematical framework and added critical ablations (e.g., RLT + AMRB) to isolate the contributions of astromorphic components. While its performance against the latest industry-scale models remains an open question, the reviewers (including Reviewer dP2F) agreed that operationalizing these biological principles into a scalable, performant architecture is a significant and solid contribution to the NeuroAI field.

**Reviewer Concerns:**

The authors adequately addressed most of the reviewers' concerns.

**Reviewer Scores:**

It is hard to tell. The authors did a thorough job in their responses.

---

### Decision · Program_Chairs · 2026-01-26

Accept (Poster)